# Rhabdomyosarcoma: Current Therapy, Challenges, and Future Approaches to Treatment Strategies

**DOI:** 10.3390/cancers15215269

**Published:** 2023-11-02

**Authors:** Ali Zarrabi, David Perrin, Mahboubeh Kavoosi, Micah Sommer, Serap Sezen, Parvaneh Mehrbod, Bhavya Bhushan, Filip Machaj, Jakub Rosik, Philip Kawalec, Saba Afifi, Seyed Mohammadreza Bolandi, Peiman Koleini, Mohsen Taheri, Tayyebeh Madrakian, Marek J. Łos, Benjamin Lindsey, Nilufer Cakir, Atefeh Zarepour, Kiavash Hushmandi, Ali Fallah, Bahattin Koc, Arezoo Khosravi, Mazaher Ahmadi, Susan Logue, Gorka Orive, Stevan Pecic, Joseph W. Gordon, Saeid Ghavami

**Affiliations:** 1Department of Biomedical Engineering, Faculty of Engineering and Natural Sciences, Istinye University, Sariyer, Istanbul 34396, Türkiye; alizarrabi@gmail.com (A.Z.); atefeh.zarepour@gmail.com (A.Z.); 2Section of Orthopaedic Surgery, Department of Surgery, University of Manitoba, Winnipeg, MB R3E 0V9, Canada; dperrin2@hsc.mb.ca (D.P.); sommerm3@myumanitoba.ca (M.S.); 3Department of Human Anatomy and Cell Science, University of Manitoba College of Medicine, Winnipeg, MB R3E 0V9, Canada; mah.kavoosi@gmail.com (M.K.); bhavya.bhushan@mail.mcgill.ca (B.B.); machajf@gmail.com (F.M.); jakubrosikjr@gmail.com (J.R.); kawalecp@myumanitoba.ca (P.K.); saba.afifi@gmail.com (S.A.); mreza.bolandi@gmail.com (S.M.B.); koleinipeiman@gmail.com (P.K.); benjamin.lindsey@umanitoba.ca (B.L.); susan.logue@umanitoba.ca (S.L.); joseph.gordon@umanitoba.ca (J.W.G.); 4Biotechnology Center, Silesian University of Technology, 8 Krzywousty St., 44-100 Gliwice, Poland; mjelos@gmail.com; 5Section of Physical Medicine and Rehabilitation, Department of Internal Medicine, University of Manitoba, Winnipeg, MB R3E 0V9, Canada; 6Faculty of Engineering and Natural Sciences, Sabanci University, Tuzla, Istanbul 34956, Türkiye; seraph@sabanciuniv.edu (S.S.); nilufercakir@sabanciuniv.edu (N.C.); bahattinkoc@sabanciuniv.edu (B.K.); 7Department of Influenza and Respiratory Viruses, Pasteur Institute of Iran, Tehran 1316943551, Iran; mehrbode@yahoo.com; 8Department of Anatomy and Cell Biology, School of Biomedical Sciences, Faculty of Science, McGill University, Montreal, QC H3A 0C7, Canada; 9Department of Physiology, Pomeranian Medical University, 70-111 Szczecin, Poland; 10Department of Biochemistry and Molecular Biology, University of Chicago, Chicago, IL 60637, USA; 11Department of Chemistry, University of Chicago, Chicago, IL 60637, USA; 12Section of Neurosurgery, Department of Surgery, University of Manitoba, Health Sciences Centre, Winnipeg, MB R3A 1R9, Canada; 13Genetics of Non-Communicable Disease Research Center, Zahedan University of Medical Sciences, Zahedan 9816743463, Iran; amirt112@yahoo.com; 14Department of Analytical Chemistry, Faculty of Chemistry, Bu-Ali Sina University, Hamedan 6517838695, Iran; madrakian@gmail.com (T.M.); ahmadi.mazaher@yahoo.com (M.A.); 15Department of Food Hygiene and Quality Control, Division of Epidemiology, Faculty of Veterinary Medicine, University of Tehran, Tehran 1419963114, Iran; houshmandi.kia7@ut.ac.ir; 16Integrated Manufacturing Technologies Research and Application Center, Sabanci University, Tuzla, Istanbul 34956, Türkiye; ali.fallah@sabanciuniv.edu; 17Sabanci University Nanotechnology Research and Application Center (SUNUM), Tuzla, Istanbul 34956, Türkiye; 18Department of Genetics and Bioengineering, Faculty of Engineering and Natural Sciences, Istanbul Okan University, Istanbul 34959, Türkiye; arezoo.khosravi@okan.edu.tr; 19NanoBioCel Research Group, School of Pharmacy, University of the Basque Country (UPV/EHU), 01007 Vitoria-Gasteiz, Spain; gorka.orive@ehu.es; 20University Institute for Regenerative Medicine and Oral Implantology–UIRMI (UPV/EHU-Fundación Eduardo Anitua), 01007 Vitoria-Gasteiz, Spain; 21Bioaraba, NanoBioCel Research Group, 01006 Vitoria-Gasteiz, Spain; 22Department of Chemistry and Biochemistry, California State University Fullerton, Fullerton, CA 92831, USA; specic@fullerton.edu; 23College of Nursing, Rady Faculty of Health Science, University of Manitoba, Winnipeg, MB R3E 0V9, Canada; 24Biology of Breathing Theme, Children Hospital Research Institute of Manitoba, University of Manitoba, Winnipeg, MB R3E 0V9, Canada; 25Autophagy Research Center, Shiraz University of Medical Sciences, Shiraz 7134845794, Iran; 26Academy of Silesia, Faculty of Medicine, Rolna 43, 40-555 Katowice, Poland; 27Research Institutes of Oncology and Hematology, Cancer Care Manitoba-University of Manitoba, Winnipeg, MB R3E 0V9, Canada

**Keywords:** alveolar rhabdomyosarcoma, apoptosis, autophagy, unfolded protein response, bioengineering, tumor stiffness, autophagy

## Abstract

**Simple Summary:**

Rhabdomyosarcoma (RMS) is a rare pediatric sarcoma affecting skeletal muscle in children and young adults. It is responsible for 3% of all childhood malignant tumors and is the third most prevalent pediatric extracranial solid tumor. Despite advances in diagnostic and treatment methods and clinical trials to improve pediatric RMS survival rates, children with high-risk RMS and recurrent disease have 5-year survival rates of less than 30% and 17%, respectively. The cure rate remains low and the current RMS therapies continue to pose potential life-threatening toxicities, which can lead to lifelong morbidity. The treatment strategies for RMS include multi-agent chemotherapies after surgical resection with or without radiotherapy. Here, we focus on chemotherapy strategies and discuss the impact of apoptosis, autophagy, and the UPR that are involved in the chemotherapy response. Then, to screen future therapeutic approaches and promote muscle regeneration, we discuss in vivo mouse and zebrafish models and in vitro three-dimensional bioengineering models.

**Abstract:**

Rhabdomyosarcoma is a rare cancer arising in skeletal muscle that typically impacts children and young adults. It is a worldwide challenge in child health as treatment outcomes for metastatic and recurrent disease still pose a major concern for both basic and clinical scientists. The treatment strategies for rhabdomyosarcoma include multi-agent chemotherapies after surgical resection with or without ionization radiotherapy. In this comprehensive review, we first provide a detailed clinical understanding of rhabdomyosarcoma including its classification and subtypes, diagnosis, and treatment strategies. Later, we focus on chemotherapy strategies for this childhood sarcoma and discuss the impact of three mechanisms that are involved in the chemotherapy response including apoptosis, macro-autophagy, and the unfolded protein response. Finally, we discuss in vivo mouse and zebrafish models and in vitro three-dimensional bioengineering models of rhabdomyosarcoma to screen future therapeutic approaches and promote muscle regeneration.

## 1. Introduction

Rhabdomyosarcoma (RMS) is a pediatric soft tissue malignancy with poor survival rates for the high-risk and recurrent disease and has the potential for significant morbidity associated with treatment. This review will characterize the clinical implications, methods of tumor differentiation, and current chemotherapeutic agents that are involved in RMS management. Furthermore, we will discuss the roles of apoptosis, autophagy, and the unfolded protein response (UPR) and their implications in RMS chemotherapy. As we focus on these cell death pathways, we will discuss the role of zebrafish and mouse models of RMS and the relevance of tissue engineering strategies in RMS, emphasizing their importance in further understanding RMS and to direct future advances in the treatment. We are optimistic that this review will provide meaningful knowledge to guide further clinical advancements in RMS therapy in order to improve survival outcomes for RMS cancer patients.

RMS, a cancer of skeletal muscle tissue, is the most common pediatric soft tissue sarcoma. RMS is responsible for 3% of all childhood malignant tumors and is the third most prevalent pediatric extracranial solid tumor [1,2]. For individuals under the age of 20, the incidence of RMS is approximately 4.5 patients per million in the United States, accounting for approximately 350 new cases each year with half of the diagnoses occurring in the patients under the age of 10 [1,2,3]. Treatment of RMS presents unique challenges when attempting local control due to the rarity of the disease and various anatomical sites in which the primary tumor can appear [3,4].

Patient survival rates depend upon several variables such as the tumor subtype, size, grade, primary site, as well as RMS disease stage and clinical group [4]. The overall 5-year survival rates for RMS in children have exceeded 70% [4,5,6,7,8,9,10,11]. However, despite advances in diagnostic and treatment methods over the past few decades, children with high-risk RMS and recurrent disease have 5-year survival rates of less than 30% and 17%, respectively [1,12,13]. Prognostic stratification is significant because 15–20% of children have diffused metastatic disease at the time of diagnosis [14,15]. Adults with RMS also experience poor outcomes, with 5-year survival rates ranging from 26.6 to 61% [16,17,18]. Over the last three decades, there have been several national and international clinical trials which have resulted in refined treatment regimens based on the tumor stage and clinical group, leading to improved pediatric RMS survival rates [4,5,6,7,8,15,19]. In addition, advancements in molecular biology and next generation sequencing have allowed researchers and clinicians to further understand RMS pathogenesis and classification [1,4]. However, despite these developments, the cure rate for pediatric patients with metastatic or recurrent disease remains low and current RMS therapies continue to pose potential life-threatening toxicities, which can lead to lifelong morbidity [4].

## 2. RMS Subtypes

RMS is generally characterized into four main tumor subtypes, as recognized by the World Health Organization (WHO): embryonal RMS (ERMS), alveolar RMS (ARMS), pleomorphic RMS (PRMS), and sclerosing/spindle RMS (Table 1) [1,3,4,20]. The most common subtypes are ERMS and ARMS, whereas spindle cell/sclerosing RMS and PRMS are considered rare. Primary RMS tumors tend to occur at three main anatomical regions including the head and neck regions (35–40%), genitourinary system (25%), and the trunk/extremities (20%) [3]. Of the RMS tumors occurring in the head and neck region, 75% arise in the orbit of the eye [3,21].

ERMS is the most common subtype (~60–70% of cases) with bimodal distribution and peak incidence in the 0–4 and 14–18 age ranges [2,22]. In patients with ERMS, the primary tumor is commonly located in the head and neck region, specifically the superior nasal quadrants and eye socket, as well as the genitourinary system, where it is most often found in the bladder and prostate [23]. ERMS is associated with a loss of heterozygosity at the 11p15 locus in 80% of the cases, which results in an altered insulin-like growth factor 2 (IGF2) gene [3,23]. Histologically, ERMS is composed of immature rhabdomyoblasts in a stroma-rich background and lacks the alveolar pattern seen in ARMS [24]. According to the Children’s Oncology Group Soft Tissue Sarcoma (COG-STS) risk stratification, low-risk ERMS has the most favorable prognosis of the RMS subtypes with a 5-year survival of approximately 80–90% [25].

ARMS is the second most common subtype of RMS that tends to occur in late childhood/adolescence [2]. The ARMS primary tumor tends to arise on the trunk and extremities, but can also be located in the inferior orbit [23]. Histologically, ARMS is characterized by densely packed, small, round cells lining septations that resemble fetal alveoli [4]. Next generation DNA and RNA sequencing has allowed us to characterize 80% of the patients with ARMS as fusion-positive (FP), with 60% containing the PAX3-FOXO1 fusion onco-protein and 20% containing PAX7-FOXO1 fusion onco-protein (Table 1) [3]. Fusion status is clinically important as fusion-negative (FN) ARMS has molecular similarities to ERMS, and clinical outcomes of children with FN ARMS are analogous to those of ERMS [26]. ARMS is considered an intermediate/high-risk RMS subtype by the COG-STS risk stratification [12]. Intermediate-risk ARMS has an estimated 5-year survival rate of 65–73%, whereas high-risk lesions have a 5-year survival rate of less than 30% [25].

**Table 1 cancers-15-05269-t001:** Characteristics of malignant rhabdomyosarcoma subtypes.

	Embryonal	Alveolar	Pleomorphic	Spindle Cell/Sclerosing ^++^
Prevalence	2.6% (most common) [2] *	1.0% (common) [2] *	Rare [2,20]	Rare [2,20]
Age	Bimodal distribution: peak incidence ages 0–4 > 14–18 [2,22]	Late childhood/adolescents [3,23]	40–70 yrs of age, peak during 6th decade of life [27]	Children [4,28]
Gender predominance	Male [4]	None	Male [3]	NA
Subtypes	Spindle cell and Botryoid subtypes	NA ^+^	Classic, round cell, and spindle cell subtypes	NA
Primary tumor location	Head/neck, superior nasal quadrants, eye socket, bladder, and prostate [23]	Trunk and extremities, inferior orbit [23]	Lower extremities [3,4]	Head/neck region, paratesticular region [4,23]
Genetics	80% have loss of heterozygosity at 11p15 (IGF-2 gene) [3]; associated with familial cancer syndromes, e.g., LFS, NF1	60% are t(2:13)(q35:114): PAX3-FOXO1 positive [3]; 20% are t(1;13)(p36;q14): PAX7-FOXO1 positive [1]; 20% are FN; resemble ERMS characteristics/prognosis [3]	NA	NA
Histology	Immature rhabdomyoblast, less dense stromal rich background vs. ARMS, lacks alveolar pattern [24]	Densely packed, small, round cells lining septations that resemble fetal alveoli [4]	Differentiated from high-grade soft tissue sarcomas by the presence of skeletal muscle proteins on immunohistochemistry [3,4]	NA

IGF-2: insulin growth factor-II, LFS: Li-Fraumeni syndrome, NF1: neurofibromatosis type 1, FFS: failure free survival, FOXO1: forkhead box protein O1, t(2:13): translocation between chromosomes 2 and 13, t(1:13): translocation between chromosomes 1 and 13, FN: fusion negative, ERMS: embryonal rhabdomyosarcoma. * Per 1,000,000 population in US. + different sources divide botryoid as subtypes of ERMS [23] vs. subtypes of ERMS and ARMS [4], ++ relatively new subtype, thus not much information is available.

PRMS primarily occurs in adults between the ages of 40–70 with a peak incidence during the 6th decade of their life. PRMS is most often found in the lower extremities and can be subdivided into classic, round cell, and spindle cell subtypes based on the histological findings [4]. Immunohistochemistry and the presence of skeletal muscle proteins are used to differentiate PRMS from other high-grade soft tissue sarcomas found in adults [3,4]. In general, individuals diagnosed with PRMS have a worse prognosis relative to those with ERMS and ARMS due to unfavorable anatomic location of the primary tumor and a higher likelihood of being treated outside of a sarcoma specialized center [3]. In addition, PRMS is unique as it does not respond to chemotherapy, unlike ERMS and ARMS, and is often treated with radiation therapy with wide excision.

## 3. RMS Classification

The classification of RMS subtypes has changed over the last several years due to the advancements in nucleic acid sequencing [1]. Initially, RMS was divided into two main subtypes, ERMS and ARMS, based on the light microscopy findings [29,30]. ARMS and ERMS both contained cells that resembled immature skeletal myoblasts distributed around an open central space [4,29,30]. RMS differs from other small round blue cell tumors such as neuroblastoma and Ewing sarcoma via immunohistochemical staining revealing muscle cell markers such as alpha-actin, MyoD1, myogenin, and desmin [3].

Recently, molecular biology approaches have further characterized RMS by the presence or absence of fusion proteins related to the balanced translocations between chromosomes 1 and 13 (t(1;13)) and chromosomes 2 and 13 (t(2;13)), which gave way to FP and FN classifications [1,4]. These fusion proteins are composed of paired box proteins PAX3 and PAX7 and forkhead box protein O1 (FOXO1), which are transcribed yielding functional PAX3-FKHR and PAX7-FKHR transcription factors. Approximately, 60% of ARMS tumors can be characterized by t(1;13) (q35;q14) and the PAX3-FOXO1 fusion protein, while 20% of the ARMS tumors are characterized by t(2;13) (p36;q14) and the PAX7-FOXO1 fusion protein [1,3,4,22,23]. When comparing the historical subtype classification by microscopy and immunohistochemistry to the FP/FN classification, several studies found that 20% of FN ARMS tumors act more similarly to ERMS with regards to its prognosis and treatment, despite their histological differences. This shift in the classification of RMS subtypes creates unique challenges when comparing the past and present literature due to the crossover in tumor subtypes between the microscopy-based classification in the early literature and the more recent FN/FP classification of ARMS [4].

## 4. RMS Epidemiology

RMS accounts for approximately 4.5% of all cases of childhood cancer [31,32], with a bimodal distribution displaying peak incidence rates at 2–6 and 10–18 years of age as well as a slight male predominance [23,33]. The incidence of RMS is similar among countries around the world with the exception of East Asia. For example, the incidence of RMS is 4.5 cases per million (<20 years of age) in the United States and 4.9 cases per million (<15 years of age) in Sweden; however, in Japan, India, and China, the incidence of RMS is 2 cases per million [23,34,35]. In the adult population, soft tissue sarcomas comprise less than 1% of all solid tumor malignancies, with RMS comprising less than 4% of adult soft tissue sarcomas [4,23,27]. From 1975 to 2020, there has been a stable incidence rate of RMS despite ongoing advances in the diagnosis and classification of RMS subtypes [36,37]. With regards to FP disease, PAX7-FOXO1 positive RMS tends to occur at a younger age than PAX3-FOXO1 positive RMS [22]. The overall risk of RMS is lower in Hispanics and is higher in those with familial cancer syndromes, particularly Li-Fraumeni syndrome (LFS) [4].

## 5. RMS Treatment

The mainstay of treatment for RMS involves multi-agent systemic chemotherapy in order to eradicate disseminated disease, along with surgical resection of the primary tumor with or without addition of the ionizing radiation therapy for the control of local disease [4,38]. The timing of systemic chemotherapy remains controversial; however, most North American centers will administer chemotherapy in the neoadjuvant setting. Surgical resection has been shown to increase survival in Group I and II diseases, whereas those with Group III disease may experience increased morbidity without improved survival rate [4,15,39,40]. Complete surgical resection with circumferential margins greater than 0.5–1 cm is the preferred treatment method for localized RMS [3,23]. Adequate negative margins are required unless the surgical excision threatens adjacent organs, leads to the loss of significant function, results in poor cosmesis, or is not technically feasible [23]. It is therefore crucial in clinical practice to carefully assess the extent of surgery, as more extensive excisions are often associated with the sacrifice of normal functions and post-operative complications that greatly affect the quality of life, whereas too limited surgery can result in local recurrence and reduced overall survival [41]. Several factors must be considered, such as the patients’ age, histological subtype of the tumor, its anatomical location, or its size. The surgical team should balance the quality of resection with the potential sacrifice of surrounding structures and organs. Generally, mutilating operations in pediatric surgery, such as orbital exenteration, total cystectomy, or pelvic exenteration, should not be considered at primary resection [42]. Those clinical dilemmas are especially evident in the case of genitourinary and perianal lesions. In the case of tumors located in the bladder, primary resection is only indicated for small tumors in the dome of the bladder that are located away from the bladder trigone. Preservation of the urinary tract drainage is crucial to minimize nephrotoxicity of chemotherapy [43]. In cases where the tumor responds well to chemotherapy, a delay in surgery may be considered to enable a more conservative approach [44]. To avoid erectile dysfunction when the surgical margin is close to the neurovascular bundle of the penile corpora, a total cystectomy with brachytherapy of the prostate may be considered as an alternative approach to radical cysto-prostatectomy [44]. Additionally, in the case of retroperitoneal lymph node dissection, nerve-sparing techniques should be used, whereby the sympathetic chains and nerve fibers are prospectively identified to minimize the damage to sexual function [45].

To ensure that the margins of resection are adequately evaluated, a close cooperation between surgeon and pathologist is warranted. Ideally, the pathologist should be provided with adequate information on the location and orientation of the taken biopsies with the fresh tissue being directly sent to the laboratory. One of the emerging new techniques that could improve resection accuracy is fluorescence-guided surgery that allows for the visualization of tumors in real-time using, most commonly, indocyanine green dye [46]. While it is investigated in the context of various types of malignancies with promising results, the data on its application in RMS are relatively limited due to the low number of RMS cases included in the studies [47].

In the event that there are positive margins, patients may undergo radiation therapy or further surgical resection of the tumor. Re-excision of recurrent RMS has been shown to increase 5-year survival rates from 8% to 37% compared to the patients without aggressive re-excision [48]. With extensive excisions being associated with greatly reduced quality of life due to loss of functions, reconstructive procedures need to be considered early in the process of planning of local therapy. In some cases, reconstruction can be performed during the same procedure or be delayed to another time. While it is advantageous to have histological evaluation before reconstructive surgery, in some cases, resection and reconstructive surgery must be performed at the same time [49,50]. Moreover, it is crucial to consider that the following radiation therapy may disturb the functioning of implants used for joint replacement; therefore, such procedures should accommodate for radiotherapy planning.

There are two treatment modalities using radiation. The first one, brachytherapy, consists of ^125^I seeds implantation inside a tumor. It delivers a high local dose to the tumor, sparing its surroundings if located precisely with ultrasound or computed tomography guidance. Moreover, brachytherapy facilitates quick recovery due to its minimal invasiveness [51]. In radiation therapy, also called radiotherapy, ionizing radiation is applied from the device outside the patient. Both treatment options use high energy on neoplastic cells to damage their DNA. These therapies were found to be curative in various malignancies, primarily if the tumors have not spread to other organs. Radiation therapy and brachytherapy are often used as adjuvant or neoadjuvant therapy. They are recommended therapeutic options after neoadjuvant chemotherapy for all IRS-III patients, including ones with radiological remission [52].

The 3-year overall survival in a group of patients with metastatic RMS receiving radical radiation therapy is 84% in comparison with 23% for those after no irradiation. Radiation therapy at the primary site improves event-free survival (EFS) [53]. Nevertheless, local therapy to all distant metastatic sites improves the 5-year progression-free and overall survival (31.3% vs. 0%; *p* = 0.002 and 37.3% vs. 0%; *p* < 0.001, respectively) [54].

Orbital RMS is associated with outstanding survival. However, surgical treatment side effects and complications, including loss of sight, are severe [55]. Brachytherapy was introduced to orbital RMS treatment to minimize those events’ frequency. The clinical effects of brachytherapy on patients with primary orbital RMS are impressive [56]. During 57 ± 17.43 months of follow-up, 90% of patients achieved remission and 10% achieved partial remission. The survival rate was 100%. Only 20% of the patients suffered from treatment side effects like loss of sight, corneal opacity, or eyeball movement disorder. In the 11.5-year follow-up, brachytherapy was an effective local treatment against RMS in children (median age—7.4 years; 0.7–16.1 years). Moreover, it was associated with fewer adverse events than radiation therapy [55].

Although aggressive local therapy with brachytherapy and radiation therapy is recommended, prospective clinical trials are strongly needed to evaluate these treatment modalities’ effectiveness in RMS [51].

Currently approved chemotherapeutic agents for the treatment of RMS include cyclophosphamide, actinomycin-D, doxorubicin, etoposide, ifosfamide, irinotecan, melphalan, temsirolimus, vincristine, and volasertib (Figure 1).

In North America, chemotherapy for the pediatric patients consists of a backbone of vincristine, actinomycin D, and cyclophosphamide (VAC) [1,21]. In comparison, in Europe, VAC therapy is substituted for a regimen consisting of ifosfamide, vincristine, and actinomycin D (IVA), which has produced similar treatment outcomes [4,5]. There is no standardized chemotherapy regimen for the adult patients, with some studies suggest using a combination of ifosfamide, doxorubicin, and vincristine, while others utilize pediatric regimens such as VAC [3,57]. When treating RMS, chemotherapy is typically administered in intervals over a 6- to 9-months period [4]. The Children’s Cancer Study Group A Trial in the 1960s and 1970s reported up to 50–60% disease recurrence in the patients who did not receive chemotherapy [58]. This finding further emphasizes the importance of chemotherapy and its contribution to a successful multimodal curative treatment regimen. Though the chemotherapeutic regimen for RMS treatment has remained unchanged over the last few decades, current research is still evaluating the efficacy of additional drugs such as doxorubicin, cisplatin, and etoposide to the VAC therapy and the impact of variable chemotherapeutic dosing intensities [3,19]. The current literature fails to show a therapeutic advantage for higher doses of cyclophosphamide in children with intermediate-risk tumors [59]. However, the COG does show improvement in the patients with disease relapse who use irinotecan and vincristine as part of their chemotherapy regimen [60].

Radiation therapy plays an important role in the treatment of patients with COG group II (microscopic residual) or group III (gross residual) diseases [3,61]. Dosing varies based on the patient’s clinical group, with patients in clinical group II typically receiving 40 Gy of radiotherapy, whereas those in group III typically receive 50 Gy [3]. The current literature is focused on balancing the effectiveness of radiotherapy in decreasing tumor size with the reduction of treatment side effects in young patients with RMS [3]. Advents such as the use of intensity modulated radiation therapy (IMRT) and proton beam therapy (PBT) are currently being used to try and achieve this goal [3]. Notable side effects of radiotherapy include joint stiffness, soft tissue changes, appendicular skeletal growth problems, and secondary malignancy.

Although the 5-year survival of the patients with low-risk disease has approached 90%, children with metastatic disease have an overall survival rate of 25–30% at 3 years, despite the use of high dose of chemotherapy and stem cell rescue treatments [4,14,62,63]. Thus, there are several agents that are currently under investigation to improve treatments for this cohort with poor survival outcomes. For example, targeted therapeutic agents such as cixutumumab, crizotinib, pazopanib, sorafenib, and temsirolimus are currently being studied for their role in RMS treatment (Table 2). There are also various ongoing clinical trials for chemotherapeutic agents such as vinorelbine and trabectedin and combination drugs such as mocetinostat and vinorelbine, dasatinib and ganitumab, and olaparib and temozolomide (Table 3). The most important chemotherapy medications for RMS are summarized in the following Sections.

### 5.1. Temsirolimus

Temsirolimus is a second-generation analog of a natural product rapamycin—a macrolide antibiotic produced by the bacterium Streptomyces hygroscopicus [72]. The ester group of temsirolimus is hydrolyzed by cytochrome P450 CYP3A4 to its active metabolite, rapamycin [73].

Temsirolimus was approved in 2007 for the treatment of advanced renal cell carcinoma. As a specific inhibitor of mammalian target of rapamycin (mTOR), it can also be used for the treatment of various tumors where mTOR is excessively activated [74]. In 2012, temsirolimus was used in a phase II study in children and adolescents with high-grade glioma, neuroblastoma, or RMS [69]. In this study, patients received temsirolimus (75 mg/m weekly) for twelve weeks, yet this treatment did not meet the primary objective efficacy threshold. However, some promising results have been published in a more recent study, where patients received temsirolimus treatment four times over a period of 21 days, together with vinorelbine and cyclophosphamide [60].

### 5.2. Vincristine

Several vinca alkaloids extracted from the leaves of Catharanthus roseus (periwinkle) are potent inhibitors of polymerization and cell division [75]. Targeting microtubules has been a promising strategy for the development of novel anticancer therapies since they play an important role in the mitosis process. Vinca alkaloids bind in the proximity of the single high-affinity site on the (+)-end of the tubules and decrease the uptake of guanosine-5’-triphosphate (GTP), which is essential for tubule elongation [76,77]. There are three currently available vinca alkaloids: vincristine, vinblastine, and vinorelbine. Vincristine, which acts by binding most tightly to the active site [78], is the least lipophilic of the three alkaloids and has the longest half-life, resulting in a greater antitumor efficacy [79,80].

Originally, vincristine was formulated as a sulfate salt that has been approved to treat acute leukemia and as a part of a multidrug regimen for Hodgkin’s and non-Hodgkin’s lymphomas. It can also be used for the treatment of gliomas, RMS, neuroblastoma, Wilms tumor, and soft tissue cancers [81,82]. Since cellular mechanisms of resistance to vinca alkaloids have been observed in the clinical applications, combination therapies with other chemotherapeutic agents are preferred over monotherapy [83]. For the treatment of RMS, vincristine is used in combination with dactinomycin or as a combination with dactinomycin and cyclophosphamide (VAC). In some cases, VAC uses alternating vincristine and irinotecan, also known as VAC/VI. Many multitarget approaches that include vincristine are currently in various phases of clinical trials (Table 3).

**Table 3 cancers-15-05269-t003:** RMS chemotherapies.

Treatment	Clinical Trial Phase	Reference
Ifosfamide/vinorelbine	III	[84]
Ifosfamide/doxorubicin	III	[85]
Vincristine, dactinomycin, and cyclophosphamide or vincristine, dactinomycin, and cyclophosphamide/vincristine and irinotecan	III	[86]
Trabectedin	II	[87]
Irinotecan or vincristine and irinotecan	II	[13]
Vincristine, doxorubicin, and cyclophosphamide/ifosfamide and etoposide	II	[88]
Vincristine, irinotecan, and temozolomide	N/A	[89]
Vincristine and irinotecan + vincristine, doxorubicin, and cyclophosphamide/ ifosfamide and etoposide + temozolomide	II	[71]
Temozolomide + irinotecan	Preclinical (mouse models)	[90]

### 5.3. Doxorubicin

Doxorubicin is a natural product that belongs to the antibiotic group of antineoplastic agents and was originally isolated from Streptomyces paucities [82,91]. These compounds target DNA function through several mechanisms, including alkylation, intercalation, and inhibiting enzymes crucial for the process of DNA replication [92]. Doxorubicin has also a quinone moiety that participates in electron-transfer reactions and makes reactive oxygen species (ROS), including singlet oxygen, hydroxyl radicals, and peroxides. ROS are known to cause damage to DNA, RNA, proteins, and lipids, which may eventually lead to cell death. This mechanism is notably responsible for the peroxidation of myocardial lipids and therefore cardiac toxicity of doxorubicin, which is the most important and severe complication [93]. Doxorubicin is extensively used in the treatment of a variety of carcinomas including breast cancer, Hodgkin’s and non-Hodgkin’s lymphomas, sarcomas, leukemia, and thyroid carcinoma [94,95].

### 5.4. Actinomycin D (Dactinomycin)

Actinomycin D also belongs to the group of antibiotic antineoplastics. It was first isolated from Streptomyces parvullus in 1940 [96]. These drugs have intercalating properties and usually contain a flat aromatic moiety capable of slipping into the double helix of DNA and distort its structure. Actinomycin D has a planar, aromatic portion, known also as actinocin or phenoxazine system (which is accountable for the yellow-red color of the drug) and two cyclic pentapeptides connected to this aromatic moiety [97]. It has also been revealed that using low doses of dactinomycin results in ribosomal stress, resulting in p53 stabilization and activation. The p53 protein is an important transcription factor that regulates multiple genes involved in cell cycle arrest, apoptosis, differentiation, and even prevention of angiogenesis. Accordingly, administration of low doses of actinomycin D in combination with other antineoplastic agents is a promising cancer therapy [98,99,100,101]. The high affinity of dactinomycin for DNA also results in a long half-life [102]. Interestingly, cancer cells that show resistance to vincristine are also resistant to dactinomycin and doxorubicin [103]. This drug is the most effective therapy in the treatment of RMS and Wilms tumor in children. It is also used in several other carcinomas, such as Kaposi sarcoma, Ewing sarcoma, gestational trophoblastic tumors, and testicular cancer [93].

### 5.5. Cyclophosphamide

Cyclophosphamide belongs to the group of alkylating agents and is a derivative of the first alkylating agent used as an anticancer therapeutic, the nitrogen mustard compound chlormethine [104]. The main characteristic of these antitumor drugs is their highly electrophilic nature and ability to form covalent bonds with nucleophilic groups present on the nucleic acid bases of DNA [105]. These agents can alkylate nucleophilic groups on non-tumor DNA and proteins as well, which leads to many adverse effects [106]. However, the fact that cancer cells divide faster than healthy cells makes these drugs strong candidates in anticancer therapeutic approaches. Cyclophosphamide is one of the most commonly used drugs in a wide variety of hematopoietic and solid tumors, some autoimmune diseases, and in bone marrow transplants, as a single agent and also in combination chemotherapy [107].

### 5.6. Ifosfamide

Ifosfamide is also a nitrogen mustard derivative that was developed as a structural isomer of cyclophosphamide in the 1960s. It is used in adults and pediatrics as a single agent or in combination with other chemotherapeutic agents in the treatment of both hematological and non-hematological disease [108]. Ifosfamide is a prodrug activated by the CYP P450 enzymatic system in the liver to form 4-hydroxyifosfamide. Ifosfamide can pass the blood–brain barrier and can therefore cause neurotoxicity ranging from mild somnolence and confusion to severe encephalopathy and coma [109,110]. Ifosfamide can also cause a greater degree of urotoxicity compared to cyclophosphamide, thus co-administration of mesna is recommended [111]. Compared to cyclophosphamide, which is taken orally, ifosfamide is given intravenously [112]. Although the oral bioavailability is very good, the oral administration of ifosfamide is linked to severe neurotoxicity [113].

### 5.7. Melphalan

Melphalan is another alkylating agent and is commonly used in the treatment of multiple myeloma [114]. The mechanism is similar to other alkylating agents; it will alkylate the guanine base of DNA (N-7 of guanine is highly nucleophilic), which will prevent normal base pairing and lead to the inhibition of replication [115]. This drug is available both orally and intravenously to treat a variety of solid cancers, including breast, colon and ovary, RMS, melanoma, neuroblastoma and Ewing sarcoma, as well as various hematologic malignancies. Although it is considered more patient-compliant, there is still a long-term risk of inducing secondary leukemia/myelodysplastic syndrome and other secondary cancers [116].

### 5.8. Etoposide

Etoposide is a semisynthetic derivative of podophyllotoxin from Podophyllum peltatum, also called the mandrake plant. Etoposide is a podophyllotoxin glycoside with a D-glucose derivative and is structurally identical to the anticancer medication teniposide, with the exception of a methyl group (teniposide contains a thienyl group) [117].

Etoposide is well known as an apoptotic pathway inducer; however, current studies show that it may also be implicated in autophagic pathways. Whether etoposide’s activation of autophagic mechanisms leads in cell death or has a pro-survival effect remains unknown [118]. Severe myelosuppression is the major but uncommon adverse reaction following administration of etoposide. Other side effects include allergic reactions, vomiting, diarrhea, bone marrow suppression, nausea, stomatitis, abdominal pain, fatigue, hypotension, peripheral neuropathy, and hair loss [93].

### 5.9. Irinotecan

Irinotecan is an analog of camptothecin (CPT), a natural product isolated from the bark and stem of Camptotheca acuminate [119]. CPT anticancer activity is linked to the inhibition of topoisomerase I. The lactone ring of CPT is extremely vulnerable to hydrolysis, and topoisomerase I is inhibited by reclosing the lactone ring, resulting in trapping of a subset of topoisomerase-1-DNA complexes and preventing relegation of the DNA strand [120,121]. Irinotecan is a prodrug that damages DNA by inhibiting topoisomerase and kills cells in the S-phase. Lethal double-strand DNA breakage and cell death arise from the development of a cleavable drug–topoisomerase I–DNA complex [93,122]. It was discovered and produced for the first time in 1983 in Japan, and it has since shown significant anticancer activity against a wide spectrum of cancers [123]. Irinotecan has shown activity against colorectal, gastric, esophageal, small-cell and non-small-cell lung cancers, lymphomas and leukemia, and central nervous system malignant gliomas [119,123]. The combination of vincristine, irinotecan, and temozolomide (VIT) is frequently used to treat adolescents and children with relapsed RMS. A recent study has demonstrated that in patients with the first relapse of RMS, VIT treatment in conjunction with sufficient local control is linked with some disease control and may be another viable alternative to give patients as salvage therapy [89]. SN-38 is metabolized and inactivated by glucuronidation to SN-38G and intrahepatic CYP450 enzymes and excreted mainly in the bile. However, SN-38G can be reactivated by β-glucuronidases to SN-38 in the intestine, which is related to the intestinal damage, mucositis and diarrhea complications, and can be restored and reabsorbed [122,124].

### 5.10. Volasertib

Volasertib (BI 6727) is a potent dihydropteridinone derivative that inhibits polo-like kinase-1 (PLK1) by acting as a small-molecule ATP-competitive kinase inhibitor [125]. It is a second-generation PLK1 inhibitor that is structurally similar to BI 2536 but has been chemically modified to increase its PLK1 activity and pharmacokinetic profile (i.e., large volume of distribution and long terminal half-life resulting in extensive penetration into the tissues and prolonged tumor exposure). PLK1 is a serine/threonine kinase that has a vital role in the cell cycle progression through mitosis as well as regulation of DNA damage checkpoints. It is overexpressed in a wide spectrum of cancers including Ewing sarcomas, medulloblastomas, non-small-cell lung cancer, breast cancer, and RMS, and its elevated level has been correlated with poor prognosis in some types of neoplasms, making it a promising target in cancer therapy. Volasertib blocks cell cycle in the prometaphase, also called polo arrest, and induces apoptosis. It inhibits PLK1 at sub-nanomolar doses (IC50 0.87 nM); however, it has also been shown to inhibit PLK2 and PLK3 at higher doses (IC50 5 and 56 nM, respectively) [126,127,128,129].

Volasertib has been clinically studied for years in various drug combinations in adult patients suffering from acute myeloid leukemia and other solid malignancies with mixed outcomes [130]. At low volasertib/BI 2536 dosages, the pre-clinical effects of volasertib in combination with vincristine in fusion-negative RMS models appear to be significant; however, the effects of fusion-positive RMS models with the volasertib/vincristine combination require additional evaluation [130]. Given the broad usage of vincristine for the treatment of newly diagnosed and relapsed RMS, as well as the likely non-overlapping toxicities of volasertib and vincristine, the combination of these two drugs appears to be practical and has immediate clinical promise in both fusion genes-negative and positive RMS. If the limited proof-of-concept clinical testing confirms dosage and activity tolerance, more intense backbone chemotherapy and/or additional targeted medicines may be studied in the future, potentially improving RMS patient outcomes [130].

Overall, the treatment of low-risk tumors is evolving in an effort to decrease the burden of treatment by reserving intensive therapy for those with high-risk or recurrent disease [4]. This approach to therapy has led to higher tumor recurrence rates in Europe (where this approach has been adopted), lower treatment-associated side effects, and unchanged RMS survival rates in cases with low-risk tumors [4,7]. Genitourinary tumors are of particular concern when considering the side effects of treatment such as enuresis and sexual dysfunction, which are associated with local disease therapy (surgery and radiotherapy) [131]. Such side effects have led patients with high-risk RMS to unsuccessfully complete therapy due to attrition in addition to unplanned dose modifications outside of the protocol guidelines [132].

## 6. Apoptosis—General Considerations

Apoptosis, or programmed cell death, is one of the major mechanisms of cell death [133,134]. This process can occur either under physiological conditions, i.e., during development and differentiation of tissues, or as a result of prolonged stress induced by the environment of the cell [135,136,137]. Apoptosis is a strictly regulated process and can be distinguished from necrosis based on the characteristic morphological changes, such as chromatin condensation, fragmentation of DNA within the nucleus, or cell shrinkage [138,139]. Apoptosis can be induced in the cells either through the intrinsic mitochondrial pathway or the extrinsic death receptor pathways [140,141,142].

The intrinsic pathway involves the activity of Bcl-2 family proteins, located in the mitochondrial outer membrane. When the balance in their activity is tipped towards cell death, pro-apoptotic Bcl-2 proteins promote mitochondrial outer membrane permeability and subsequent release of cytochrome c, caspase activation, and apoptosis [143,144,145]. Conversely, the extrinsic pathway relies on stimulation of death receptors, such as Fas or TNFR (tumor necrosis factor receptor), by ligands [146,147,148]. This process is followed by the recruitment of adaptor proteins and initiator caspases—caspase 8 and 10, which form the death-inducing signaling complex [149]. An overview of apoptosis signaling pathways is illustrated in Figure 2.

### 6.1. Avoidance of Apoptosis by RMS Cells

Resistance to the programmed cell death, which allows for a proliferative advantage, is a characteristic feature of the malignant cells. Defects in apoptosis often result in resistance to the cytotoxic therapies, as current conventional treatment relies on the neoplastic cells’ ability to undergo cell death in response to toxicity [150]. In most cancers, the avoidance of cell death occurs predominantly due to the overexpression of anti-apoptotic genes or down-regulation of pro-apoptotic genes [151,152].

In fusion-positive (FP) RMS, the PAX3-FOXO1 and PAX7-FOXO1 fusion proteins function as drivers of oncogenesis by dysregulating multiple crucial cellular pathways. The fusion proteins drive the expression of other transcription factors such as MYCN and MYOD1, contributing to the RMS formation and progression [153]. Moreover, the fusion proteins drive the expression of receptor tyrosine kinases (RTKs). The overexpression or activation mutations of both genes encoding the RTKs or their downstream signaling effector genes are common in FP RMS [153]. This includes FGFR4 (fibroblast growth factor receptor 4), whose activating mutations are present in 7% of FP RMS patients, triggering RAS and STAT signaling pathways that induce tumor growth [154]. Activation of the Ras/Raf/MEK/ERK and JAK/STAT pathways can result in prevention of apoptosis through phosphorylation of Bim and Bad, which result in the loss of the ability to heterodimerize with survival proteins BCL-XL and BCL-2. Moreover, the JAK/STAT signaling pathway can result in the overexpression of anti-apoptotic BCL-XL [155,156]. Taken together, these changes result in down-regulation of BAX/BAK effector proteins and apoptosis restriction. Additionally, the overexpression of FGFR4 tyrosine kinase in RMS cell lines induces its auto-phosphorylation and constitutive signaling that result in the prevention of apoptosis by targeting the IGF1R-PI3K-mTOR (insulin growth factor 1 receptor/phosphoinositide 3-kinase/mammalian target of rapamycin) pathway [157,158,159]. Additionally, knockdown of FGFR4 in RMS cell lines shows reduction in cell proliferation and increase in apoptosis [160].

PDGFR (platelet-derived growth factor receptor) is another RTK driven by the fusion protein. Experimental data suggest that its overexpression regulates cancer cell stemness, differentiation, and apoptosis, with PDGFR inhibition resulting in an increase in apoptosis accompanied by the G2/M cell cycle arrest in RMS cell lines [161].

Other RTKs induced by the fusion protein and implicated in RMS progression can signal through the RAS-PI3K-AKT-mTOR and RAS-RAF-MAPK pathways [162,163]. Gene expression analyses reveal that over 50% of the patients with FP RMS carry mutations that impact the aforementioned pathways [164,165]. AKT serves as a member of the pro-survival pathway, as its activity rescues cells from PTEN-mediated apoptosis [166]. The anti-apoptotic activity of AKT seems to be multifactorial, as it directly phosphorylates selected components of the apoptotic machinery. Phosphorylation of BAD by AKT prevents its dimerization with a member of the BCL-2 family—BCL-XL, restoring the latter anti-apoptotic function [167]. Moreover, through direct phosphorylation, AKT inhibits the activity of caspase 9 [168]. Finally, PAX-FOXO1 fusion protein can synergize with the loss of the cyclin-dependent kinase inhibitor 2A (CDKN2A) or p53, functionally either indirectly through CDKN2A tumor suppressor gene loss or TP53 promoter mutation [169].

An increasing body of evidence suggests that epigenetic regulation contributes to RMS development and progression [170]. In comparison with normal tissue, muscle-specific microRNAs (miRs) are down-regulated. These miRs are often involved in protecting the organism from malignant transformation, serving as antioncogenes. The inhibition of these specific miRs, such as miR-29, miR-450b-5p, miR-203, and miR-214, contributes to the enhanced tumorigenesis through diminished myogenic differentiation and inhibition of apoptosis [171,172]. While those miRs affect diverse molecular pathways, the effect is partly mediated by the IGF1/AKT pathway, as transient transfection of miR-378a-3p in ARMS cell line induced apoptosis, impaired migration, and promoted myogenic differentiation [173].

### 6.2. Antineoplastic Agents Targeting the Apoptosis Pathway in RMS

There are several chemotherapeutic drugs that have been approved for RMS treatment, which act through inducing cancer cell apoptosis. These treatment modalities initiate the cell death pathway through diverse molecular mechanisms (e.g., through cell cycle blockade, interference with proliferation, or DNA damage) (Figure 2).

Alkylating agents, cyclophosphamide and ifosfamide, induce crosslinking between DNA strands (see Section 5 above). In the cell lines exposed to alkylating agents, a decrease in the DNA strand expression of the anti-apoptotic BCL-2 and an increase in the pro-apoptotic BAX, caspase 3, and PARP expression have been observed [174,175]. Moreover, a dose-dependent inhibition of ERK1/2 and AKT phosphorylation was observed, suggesting that the changes in apoptosis-associated proteins is mediated by ERK/MAPK and PI3K/AKT signaling pathways [174].

While the exact mechanism through which etoposide leads to apoptosis is not fully understood, it seems to involve AKT regulation, whereby etoposide stimulates AKT to migrate into the mitochondria, enhancing its interaction with Smac, phosphorylating it at residue 67, which in turns leads to the enhancement of Smac interaction with X-chromosome linked IAP (XIAP) protein, which then upregulates the activity of caspase 3 [176,177]. During therapy with topoisomerase II inhibitor, caspase 2, 3, and 9 activation is observed, an effect which is partly mediated by BCL-2 [178]. Similarly, treatment with dactinomycin results in apoptosis in both a caspase-dependent and independent manner. Dactinomycin treatment results in cell death through the activation of caspase 7 and 9, an affect which was only partly attenuated by caspase inhibition, suggesting the partial involvement of reactive oxygen species release and upregulation of the apoptotic-inducing factor (AIF) expression [179].

As mentioned in Section 5 above, vincristine destabilizes microtubules through suppression of tubulin polymerization [180]. As a result, cells undergo arrest in the G2/M phase. Vincristine treatment also depletes the mitochondrial membrane potential, increasing the release of mitochondrial cytochrome c into the cytosol. Additionally, there is an observable increase in tBID, which in combination with lower concentrations of BCL-2 and BCL-XL, leads to the apoptosis through FADD-associated auto-cleavage and activation of procaspase-8 [180]. Another established mechanism through which the cell cycle becomes halted in the G2/M phase is the inhibition of pro-survival polo-like kinases (PLKs) [181]. Volasertib, a novel PLK inhibitor, induces apoptosis through caspase 3 activation [181].

Melphalan and temozolomide induce apoptosis in cancer cells through distinct molecular pathways. The former induces the cleavage of MCL-1, disrupting the MCL-1/BIM complex, which under normal conditions neutralizes the pro-apoptotic function of BIM and prevents the activation of death effectors [182]. The disappearance of MCL-1 allows for the release of BIM isoforms, which lead to further BAX activation and cytochrome c release. The mechanism through which temozolomide induces apoptosis remains unelucidated, but it likely does not involve changes in MCL-1, BCL-2, BCL-XL, or BAX protein expression [182,183].

Irinotecan, a DNA topoisomerase I inhibitor increases intracellular BAX concentration. Moreover, it causes an increase in p53 and caspase 9 levels with accompanying decrease in the expression of BCL-XL [184]. Elevated p53 reinforces the induction of apoptosis by raising the expression of pro-apoptotic members of the BCL-2 family and death receptors [185]. Nevertheless, a notable subset of patients exists for whom the aforementioned treatment is ineffective by the means of rapidly acquired resistance. Therefore, significant efforts are placed to identify other, more efficacious therapeutic agents.

Temsirolimus is a derivative and prodrug of the widely used immunosuppressant sirolimus, also known as rapamycin (see Section 5 for more details). Rapamycin and its derivatives act by inhibiting mTOR [186,187]. Blockage of this protein is followed by dysregulation of proliferation and hindrance of the cell growth [187]. Moreover, mTOR inhibition leads to cell cycle arrest in the G1-phase and directs the cell towards apoptotic cell death [186]. This observation could be explained by a decrease in the mTOR downstream target p70S6K, which normally phosphorylates the pro-apoptotic BAD on serine 136, disrupting its ability to bind to BCL-XL and BCL-2 [188]. In RMS cells, mTOR inhibition can successfully abrogate tumor growth with a reduction in proliferation and invasiveness, as well as an induction of apoptosis through inhibition of BCL-2 expression [189]. The restriction of tumor growth is associated with the down-regulation of mTOR and Hedgehog (Hh) signaling, both of which are implicated in the pathogenesis of RMS. This implementation of molecular targeted therapy opens new avenues for personalized therapy in the hope to improve therapeutic outcomes [189].

The addition of temsirolimus to the chemotherapeutic regimen is expected to enhance its efficacy, as mTOR inhibition presumably resensitizes previously chemoresistant cancer cells [190]. The clinical trial (NCT00106353) reported that this agent at the dose of 75 mg/m^2^/week prolongs stable disease. However, further evaluation of temsirolimus in combination with currently used therapy regimens is essential [69].

Some other examples of the molecular targeted therapies in RMS involve the use of vascular endothelial growth factor (VEGF) inhibitors. The expression of VEGF is indicative of poor prognosis in various solid tumors, including both ARMS and ERMS [191]. These observations strongly suggest that VEGF could be a suitable therapeutic target. The clinical trial NCT01222715 compared the efficacy of temsirolimus and VEGF-A inhibitor bevacizumab, where 87 patients received the standard chemotherapy combined with one of the aforementioned agents. Temsirolimus was found to be more efficacious in terms of event-free survival between the two groups [60].

One of the targeted therapies that is currently being investigated in the context of RMS treatment (see Table 4) involves the use of IgG1 monoclonal antibody, cixutumumab, which is directed against the human insulin-like growth factor-1 receptor (IGF-1R) [192]. This therapeutic agent down-regulates the PI3K and MAP signaling pathways, increasing caspase 3 and PARP cleavage [193]. The limited activity and acceptable toxicity of monotherapy supports the idea of including this antibody in the combined therapeutic regimens [70,71,194]. Other drug combinations involving cixutumumab with doxorubicin and temsirolimus are under scrutiny [194]. The preliminary results suggest that this antibody improves the outcomes of temsirolimus therapy [195]; however, the dependence of combined therapy on IGF-1R expression in cancer cells remains unclear [195,196].

Crizotinib and ceritinib, ALK (anaplastic lymphoma kinase) and ROS1 (c-ros oncogene 1) inhibitors, are other neoplastic drugs whose efficacy against RMS is under investigation [197,198]. ALK inhibition is a known mechanism for inducing apoptosis [199], and cancer cells (such as non-small-cell lung cancers (NSCLCs) and RMS) are often dependent on ALK and ROS1 function, providing a reasonable rationale for evaluating crizotinib and ceritinib in these cancers [200]. Nevertheless, studies characterizing the properties of ALK and ROS1 inhibitors failed to prove their efficacy as single agents against RMS [68,201,202]. However, the addition of ceritinib to another chemotherapeutic agent, especially kinase inhibitors such as dasatinib or sorafenib [66], improves the therapeutic outcomes [201,203]. Sorafenib in combination with PLK inhibitors is under scrutiny in other types of cancers and primary results are promising [181]. Another kinase inhibitor pazopanib is also under investigation [204]. According to the recent studies, it seems to be a promising therapeutic modality for patients with refractory and relapsed sarcomas [64,65]. Similarly, regorafenib does not improve progression-free survival in the treatment-refractory liposarcoma [205], but its combination with other agents in RMS treatment might lead to superior results.

Trabectedin, which inhibits gene activation and blocks nucleotide excision repair, leads to cell cycle arrest [206] and upregulation of BAX, BID, and caspase 3 transcripts [207]. While this agent failed to demonstrate sufficient activity as a single agent, it might become an element of a potent multidrug regimen [87].

Another approach to influence the RMS apoptotic pathways is through inhibition of histone deacetylation. Posttranslational modifications of histones affect gene expression. Acetylation, as one of these modifications, marks regions of the high transcriptional activity [208]. However, acetylation can be reversed by the histone deacetylases (HDACs) leading to transcriptional repression. HDACs silence apoptosis inducers or tumor suppressor genes, contributing to oncogenesis [209,210]. HDAC inhibitors (HDACIs) are a promising group of therapeutic agents that are believed to restore physiological histone acetylation [209]. A study on HDACIs’ influence on apoptosis of RMS cells revealed that they, especially in combination with bromodomain and extra terminal (BET) inhibitors, trigger the mitochondrial pathway of apoptosis [211,212]. Further analysis shows that BIM and BIF become upregulated, while BCL-XL and survivin are down-regulated [211,212,213,214]. HDACIs are also capable of cell cycle arrest in the M-phase [213]. One member of the HDACIs, entinostat, demonstrates synergistic antitumor activity if combined with vincristine [215], while another HDACI SAHA acts synergistically with doxorubicin [214] and has beneficial properties against RMS in both cell and mouse models [215].

The involvement of PARP proteins in DNA repair prompted researchers to hypothesize that DNA breaks induced by radiotherapy would be more deadly to cancer cells if the therapy was combined with olaparib, a PARP1-3 inhibitor [216,217]. This hypothesis was verified in a study on RMS cells, where combined exposure to PARP inhibitors and ionizing radiation elicited more robust cytotoxic effects than radiation alone [218].

Statins or 3-hydroxy-3-methyl-glutaryl-CoA reductase (HMG-CoA reductase) inhibitors are another group of agents whose antiproliferative properties might be beneficial during anti-RMS treatments. Statins are believed to activate the mitochondrial pathway of apoptosis. Recent research suggests that simvastatin activates caspases 3 and 9 [219]. Moreover, pretreatment with statins augments pro-apoptotic properties of other antineoplastic agents [219]. A summary of the available data on HMG-CoA reductase inhibitors suggests that impairing the Ras family GTPase signaling is crucial for the chemo-sensitizing effect [220,221,222,223]. Nevertheless, the clinical significance of statins as antineoplastic agents is still undetermined [220].

## 7. Autophagy Process

Autophagy is a Greek term that means self-digestion and was firstly proposed by Christian de Duve in 1963 [224]. In vivo, basal autophagy is constitutively active under normal conditions, and it can be further induced by physiological and environmental stressors such as DNA damage, reactive oxygen species (ROS), hypoxia, nutrient starvation, endoplasmic reticulum stress, adenosine triphosphate (ATP) deficiency, hormonal stimulation, and pharmacological treatment [225,226,227,228]. Based on the mechanism and morphology, autophagy is divided into three major types: microautophagy, chaperon-mediated autophagy (CMA), and macroautophagy [229]. In microautophagy, small cargo substrates are directly delivered to the lysosome membrane without an autophagosome (a kind of double-membrane vesicle) and these substrates are digested by lysosome [230,231,232]. This type of autophagy cannot be activated by stress or nutrient deprivation (Figure 3A) [233]. CMA has the most selective function and identifies unfolded substrate proteins containing a special recognition motif KFERQ (Lys-Phe-Glu-Arg-Gln) by chaperone proteins like HSP70 and HSPA8. LAMP2A (lysosomal CMA receptor) identifies these complexes containing chaperone and substrate proteins and transports them to the lysosome [234,235,236]. Like microautophagy, during CMA, cytosolic components are not enveloped by a cytoplasmic membrane (Figure 3B) [237,238,239,240,241]. Macroautophagy is considered common autophagy [234]. This type of autophagy is evolutionarily conserved from yeast to mammals [234,242] and is defined by a process where cytoplasmic substrates are isolated by an autophagosome and then transported to the lysosome for digestion [221,234,243]. Generally, this process could be either selective or non-selective [244]. Although both selective and non-selective autophagy use the same mechanism for digesting substrates, in selective autophagy, special substrates such as mitochondria, ribosome, and peroxisomes are targeted by autophagy receptors, which contain an ATG8-interacting motif (AIM)/LC3-interacting region (LIR), to facilitate delivery to the autophagosome (Figure 3C) [245,246].

### Targeting Autophagy to Increase the Effectiveness of Chemotherapy in Rhabdomyosarcoma

Autophagy has been observed to play both a survival role and a mode of cell death in cancer cells. This dual role of autophagy in cancer development has led to two different treatment strategies. The first approach involves the sensitization of cancer cells to chemo/radiotherapy through inhibition of the cytoprotective role of autophagy, while the second approach involves the induction of autophagic cell death in apoptosis-resistant cancer cells [247]. In this regard, combinatorial therapeutic strategies produce a synergistic or additive effect compared to monotherapies to overcome the resistance of tumor cells to cancer chemotherapeutic agents and enhance their response to anticancer compounds [248]. Kwan-Hwa Chi proved that the combination of chemotherapeutic drugs with autophagy inducers and autophagy inhibitors could provoke more effective autophagic perturbations. The triplet drug combination of chloroquine (autophagy inhibitor), rapamycin (autophagy inducer), and vinorelbine (chemotherapeutic drug) produced a high synergism and inhibited cell proliferation in hepatocellular carcinoma cell lines, Huh7.5.1 and HA22T [249]. A Rezaei Moghadam and colleagues have shown that autophagy inhibition increased temozolomide (TMZ)-induced extrinsic apoptosis in ARMS cell lines. Indeed, TMZ can activate autophagy flux in ARMS cells by increasing the expression levels of autophagy proteins LC3-II, P62, and ATG5-12. Treatment of the ARMS cell line RH30 with the autophagy inhibitor Bafilomycin A1 significantly increased the antitumor effect of TMZ as a chemotherapy agent [250].

Statins are FDA-approved mevalonate (MEV) cascade inhibitors, more commonly known as cholesterol-lowering drugs, and are widely used for the primary and secondary prevention of coronary artery disease [251,252,253]. Recent studies indicate that statins also mediate the inhibition of small Rho GTPases and the regulation of GDP/GTP exchange [222,251,252,254,255,256,257,258]. Furthermore, depletion of Rho GTPase can potentially promote the induction of tumor cell apoptosis [251,259,260]. Statins induce cell death in treated cells by targeting MEV cascade-independent mechanisms [261,262]. Importantly, autophagy is induced and modulated in statin-induced cell death [263,264]. Shahla Shojaei et al. demonstrated that simvastatin enhances TMZ-induced apoptosis in human glioblastoma (GBM) cell lines. This investigation showed that simvastatin inhibited the TMZ-induced autophagic flux by blocking the fusion of autophagosomes and lysosomes [141]. Moreover, Werner et al. demonstrated that the combined application of doxorubicin and simvastatin had additive effects on activating the mitochondrial pathway of apoptosis in RMS cells compared to either drug alone [219]. Doxorubicin is a alkylating agent; however, its use is limited in the treatment of RMS due to the risk of cardiac toxicity [265]. Some studies have concluded that doxorubicin upregulates cardiac autophagy and contributes to the pathogenesis of doxorubicin-induced toxicity in vitro and in vivo [266,267,268,269]. This antitumor antibiotic damages DNA by increasing the level of ROS, which not only ceases DNA synthesis but also impedes cancer cell proliferation and induces apoptotic cell death [270,271]. Also, doxorubicin could mediate apoptosis via inducing AMP-activated protein kinase (AMPK), which triggers p53 activation [272]. Doxorubicin might also promote cytoprotective autophagy as a result of DNA damage by activating poly (ADP-ribose) polymerase-1 (PARP-1) [273]. On the other hand, doxorubicin induces autophagy via depletion of the GATA4 transcription factor and/or activation of ribosomal protein S6 kinase 1 (S6K1), which may contribute to the modulation of autophagy-associated genes such as ATG12, ATG5, Beclin-1, BCL-2, and others [274].

Vincristine is a microtubule targeting agent that arrests the cell cycle by binding to the Vinca domain in the h-tubulin subunit and promotes apoptosis [270,275]. This drug is used in several cancers including rhabdomyosarcoma [5,270,276,277], non-small-cell lung cancer [5,278], breast cancer [5,279], lymphomas [280,281], and malignant brain tumors [282,283,284]. Actinomycin D is an antibiotic that also has antitumor activity against malignancies, especially in RMS [5,276,277]. Yu Wang and his colleagues in their study depicted that the anti-Fas death receptor antibody/actinomycin D (AF/AD) induced apoptosis and P38MAPK-mediated protective autophagy in human hepatocellular carcinoma Bel-7402 cells. They showed that adding the P38MAPK inhibitor SB203580 or the autophagy inhibitor 3-methyladenine (3-MA) to this combination could induce apoptosis in Bel-7402 cells [285].

Glutathione S-transferase P1 (GSTP1), a phase II detoxifying enzyme, is overexpressed in the tumor cells and contributes to multidrug resistance (MDR). Overexpression of GSTP1 triggers autophagy through interacting with the p110α subunit of phosphatidylinositol-3-kinase (PI3K) and subsequently inhibits the PI3K/AKT/mTOR signaling to protect human breast cancer cells from adriamycin (ADR)-induced cell death. X Dong et al. indicated that GSTP1 knockdown in ADR-resistant MCF-7 human breast cancer cell lines combined with autophagy inhibition significantly reduced the resistance of MCF-7/ADR cells to ADR [286]. Multidrug resistance in pediatric rhabdomyosarcoma is also associated with the GST family of genes. A combination of GST protein inhibitors OZO-H [4-phenyl-1,3,2-oxathiazolylium-5-oleate) or etacrinic acid and cytotoxic drugs vincristine, doxorubicin, and topotecan modulates the drug sensitivity of alveolar rhabdomyosarcoma RH30 cells and embryonal rhabdomyosarcoma A204 cells and provides a noticeable additive effect on cell death [287]. This suggests a positive correlation between GST protein expression and soft tissue sarcoma resistance to adriamycin, cisplatin, and mitomycin C [288].

Autophagy function is dependent on lysosomal activity [289]; thus, inhibiting or modulating lysosomal activity could be a high-value target to improve chemosensitivity of RMS cells. Salerno et al. proved that blocking lysosomal acidification by the V-ATPase inhibitor omeprazole, or by specific siRNA, considerably potentiated the cytotoxic effects of doxorubicin against an embryonal rhabdomyosarcoma cell line, but also mitigated the invasive potential of rhabdomyosarcoma cancer stem cells [290]. Ciclopirox olamine (CPX), a synthetic hydroxypyridone derivative, is known to induce cell death in different cancer types including leukemia, breast cancer, and soft tissue sarcoma. Hongyu Zhou et al. found that CPX activates ROS-mediated JNK signaling pathway to provoke autophagy in human rhabdomyosarcoma (RH30 and RD) cells, and that inhibition of this autophagy response by chloroquine (CQ) exacerbates the anticancer effectiveness of CPX [291].

The ubiquitin-proteasome system (UPS) and the heat shock response (HSR) are two essential regulators for cell homeostasis, as their inhibition has a great impact on the growth and survival of normal cells as well as the stress response and invasion of cancer cells. Peron et al. demonstrate that a combination of a lysosomal inhibitor (chloroquine), a proteasome inhibitor (bortezomib), and a competitive Hsp90 inhibitor (17-DMAG) sensitizes the alveolar and embryonal rhabdomyosarcoma cell lines (RH30 and RD) to anticancer drug-induced apoptosis [292]. Moreover, this sensitivity could be abrogated by the autophagy activator rapamycin, confirming that autophagy is a key resistance mechanism in RMS cells [292]. In agreement with these observations, disruption of autophagosome formation via a shRNA sequence against ATG7 (shATG7) or by inhibition of both V-ATPase-dependent acidification and autophagosome–lysosome fusion using bafilomycin A1 can mitigate antitumor drug-induced autophagy and abolish the growth of embryonal (RD) and alveolar (RMS13) RMS cell lines [293].

SIRT1 and SIRT2 are deacetylase enzymes that belong to the mammalian Sirtuin (SIRT) family and are involved in various cellular processes such as metabolism [294], cell survival [295], differentiation [296], DNA repair [297], and pathogenesis of solid tumors and leukemias [298,299,300]. A study demonstrated that overexpression of SIRT1 and SIRT2 induced autophagic flux in human soft tissue sarcoma cell lines. Ma et al. further showed that pharmacological inhibition of Sirtuins with Tenovin-6 (Tv6) induced apoptosis and impaired autophagic flux in pediatric sarcoma cell lines, without impacting p53 acetylation. They indicated that using Tv6 or SIRT1 and SIRT2 siRNAs not only has antiproliferative effects in the rhabdomyosarcoma cell lines (RD and RH30), but also an anti-expression effect on the protein level of LC3-II [301].

Temsirolimus is a specific pharmacological inhibitor of mTOR that has been well tolerated by patients with advanced solid tumors and melanoma in clinical phase I trial of the combinatorial therapy. This clinical study indicates that the combination of temsirolimus and hydroxychloroquine, as autophagy inhibitor, regulates autophagy in patients, and produces more synergistic antitumor activity [302].

The adenosine triphosphate (ATP)-binding cassette (ABC) transporters consists of a large superfamily of membrane proteins that transport substrates across membranes by hydrolyzing ATP [303]. The ABCC subfamily constitutes 12 transporters and overexpression of these proteins causes chemotherapeutic drug resistance in tumor cells [304]. Among them, P-glycoprotein (P-gp/MDR1/ABCB1) is associated with resistance to commonly used chemotherapeutic agents in rhabdomyosarcoma [270]. Noticeably, many anticancer chemotherapeutics including doxorubincin [305,306], actinomycin-D [306,307], paclitaxel [308,309], etoposide [306,310], mitoxantrone [311,312], vincristine [313,314], vinblastine [315], arsenic [316,317], romidepsin [318,319], colchicine [320,321], topotecan [322], and irinotecan [323,324,325] are P-gp substrates. Chemotherapy resistance can be seen in P-gp-overexpressing cancer cells attributed to the pumping of drugs out of cells [270,304,326]. Espelt et al. have demonstrated that the inhibition of ABC transporter gene expression and autophagy-related gene expression by certain microRNAs (miRNAs) could lessen the survival rate of hepatocellular carcinoma cells under chemotherapeutic drug treatment [327]. Similarly, the inhibition of P-gp with different concentrations of silibinin di-hemisuccinate (SDH), a flavonoid antioxidant, enhanced MTX-induced cytotoxicity in MTX-resistant human rhabdomyosarcoma (hRD) [328]. The cancer stem cells (CSC) in glioblastoma multiforme (GBM) display high levels of ABC transporters, which are associated with the chemoresistance phenotype in GBM CSCs.

Polo-like kinase 1 (PLK1), a key mitotic regulator, is frequently overexpressed in multiple human cancers. A Pandey showed that the cotreatment of cells with PLK1 inhibitor volasertib, temozolomide, and radiation can overcome resistance in radio/chemoresistance of GBM, both in vitro and in vivo [329]. PLK1 is a serine/threonine kinase and is the main driver of DNA replication after stress. High PLK1 expression has been associated with poor prognosis in several cancers, including rhabdomyosarcoma. Intriguingly, overexpression of PLK1 promotes the proliferation of tumor cells through crosstalk with autophagy. Wu et al. showed that knockdown of PLK1 by shRNA could suppress proliferation and invasion of U87 and U251 glioma cells, along with increased cell apoptosis and a reduction in the expression of MMP9, ATG5, and LC3-II autophagic factors [330]. Moreover, in high-risk FP RMS, PLK1 phosphorylates PAX3-FOXO1 at Ser-503, leading to protein stabilization. Moreover, preclinical studies using the combination of PLK1 inhibitors, such as volasertib, along with other agents, such as etoposide, vincristine, vinblastine, vinorelbine, or eribulin, is widely beneficial in treating young patients with rhabdomyosarcoma and other cancers [331,332,333].

A schematic overview of autophagy targeting in relation to RMS has been shown in Figure 4.

Collectively, these studies strongly suggest that specific and effective autophagy modulators could be a beneficial adjunct in combination cancer therapy. We summarized the available data collected from previous studies about the synergistic effect of autophagy inhibitors and other therapeutic agents on RMS in Table 5.

## 8. General Concepts of Unfolded Protein Response and Its Link to RMS

The endoplasmic reticulum (ER) is the cell ‘manufacturing and packaging plant’, playing important roles in the production, folding, and post-transitional modification of proteins and biosynthesis of lipids. Given the importance of ER function, cells must constantly monitor ER health. Three ER-anchored transmembrane receptors, inositol requiring enzyme 1α (IRE1α), protein kinase R like endoplasmic reticulum kinase (PERK), and activating transcription factor 6 (ATF6), survey the internal ER environment. Under non-stress conditions, each of these receptors is inactivated through binding of their N-terminus to the ER chaperone glucose regulated protein 78 (Grp78) [334,335]. Accumulation of unfolded or misfolded proteins within the ER lumen, a condition known as ER stress, instigates Grp78 dissociation, facilitating receptor activation [334,335]. IRE1α dimerizes and trans-autophosphorylates, facilitating activation of its RNase activity [336,337,338]. Similar to IRE1α, PERK dimerizes and trans-autophoshorylates upon loss of Grp78 binding, allowing it to acquire full catalytic activity [339]. In contrast to IRE1α and PERK, upon Grp78 dissociation, ATF6 translocates to the golgi apparatus where it is cleaved by Site 1 and Site 2 proteases forming ATF6N [340]. The collective signaling pathways downstream of IRE1α, PERK, and ATF6 constitute the UPR. These pathways work in a cooperative, complimentary fashion to reduce the levels of unfolded proteins, thereby restoring ER homeostasis [341].

IRE1α via its RNase activity splices XBP1 mRNA, which following relegation by RTCB and translation produces a transcription factor referred to as spliced XBP1 or XBP1s [342,343]. XBP1s increases expression of genes encoding ER chaperone proteins and components of the ER-associated degradation machinery (ERAD) [344]. By doing so, IRE1 signaling helps to support the folding of those proteins that can be refolded while promoting the destruction of those proteins beyond repair. IRE1α RNase activity has also been linked to the degradation of selective mRNAs via a process referred to as regulated IRE1 dependent decay (RIDD) [345,346]. Many mRNAs identified as RIDD targets encode ER-targeted proteins. By facilitating their degradation, IRE1α avoids additional pressure being placed on an already stressed ER.

Similar to IRE1-RIDD signaling, PERK activation aids the resolution of ER stress by halting canonical cap dependent translation. PERK, via its kinase activity, phosphorylates Ser51 on eif2α [347]. Phosphorylation of eIF2α at Ser 51 blocks eIF2B-mediated exchange of GDP for GTP, thereby halting 5” cap-dependent translation. This translational block, while widespread, is not complete as genes with an upstream open reading frame or an internal ribosome entry site within their 5′ untranslated region (UTR) are selectively translated under these conditions [348]. Activating transcription factor 4 (ATF4) is one such example. ATF4 expression during ER stress is linked to the regulation of adaptive genes including those involved in regulating oxidative stress, amino acid metabolism, and ER chaperones [349,350]. PERK can also target and phosphorylate the transcription factor nuclear factor erythroid 2-related factor 2 (NRF2) [351]. Normally, NRF2 is retained in the cytoplasm through binding to Kelch like-ECH-associated protein 1 (KEAP1) [352]. PERK-mediated phosphorylation of NRF2 breaks the NRF2/KEAP1 interaction, enabling nuclear translocation of NRF2, where it increases the expression of pro-survival genes [351]. The ATF6 signaling pathway both upregulates expression of genes encoding ER chaperones and supports IRE1 signaling by transcriptionally upregulating XBP1, thereby ensuring a plentiful pool of XBP1 for the IRE1-mediated splicing [342].

In addition to controlling the UPR directly, IRE1, PERK, and ATF6 can stimulate and influence proteostasis through additional stress-induced pathways, in particular autophagy. Although a basally active process, levels of autophagy tend to increase during times of stress. IRE1, PERK, and ATF6 signaling pathways have all been linked to events such as upregulation of autophagy-related genes (ATG), repression of autophagy suppressive pathways such as mTORC1 signaling and disruption of Beclin 1/BCL-2 complexes, which stimulate autophagy [353].

While the UPR is an adaptive process, unlike autophagy, it is not meant to be constitutively activated. If ER stress is excessive or prolonged, UPR signaling transitions from a pro-survival to a pro-death pathway. Although ER stress-induced cell death and the mechanisms facilitating it have been extensively studied, exactly how and when a cell makes the decision to transition to death is still a matter of much debate. Regulation of BCL-2 family members leading to mitochondrial-mediated apoptosis was thought to be the predominant cell death pathway during ER stress, but recent studies have indicated a role for death receptors, in particular the trail death receptor DR5 [354].

The fundamental role of UPR signaling in healthy cells is to provide cells with means to survive during transient stress, but the situation in diseased cells, such as cancer cells, is much more complex. Unlike healthy cells, cancer cells have acquired the ability to sustain permanent activation of UPR mediators. Constitutive activation of IRE1 and PERK and their associated downstream pathways has been reported in multiple cancers including triple negative breast cancer, prostate cancer, lung cancer, and, more recently, sarcomas including RMS (RMS) [355,356,357]. Sustained UPR signaling offers cancer cells a means to meet the protein folding demands instigated by the activation of oncogenes or loss of tumor suppressors. However, the impact of UPR signaling appears to be more extensive than simply aiding ER function, with IRE1- and PERK-mediated pathways implicated in a range of pro-tumorigenic processes ranging from supporting metastasis to the development of chemoresistance [355].

Engaging pathways such as UPR, heat shock proteins, and autophagy are strategies cancer cells can exploit to maintain proteome integrity. Similar to many other cancers, RMS cells are known to have a high dependence on proteostatic pathways including the UPR [163,357]. This presents a therapeutic opportunity; if these pathways can be impeded and proteostasis disrupted, cancer cells may engage death pathways. In RMS, chemical inhibitors of heat shock protein HSP70 have been shown to reduce RMS cell viability. The addition of MAL3-101 (HSP70 inhibitor) to RMS cell lines triggers cell death via a mechanism dependent upon UPR-mediated induction of the pro-apoptotic transcription factor CHOP [358]. Subsequent studies demonstrated that while HSP70-based inhibition could elevate UPR signaling and cell death, RMS cells by increasing autophagy or ER-associated degradation pathways developed resistance [359]. Combination with strategies to decrease autophagy, such as chloroquine addition, overcame HSP70 inhibitor resistance, suggesting dual targeting of HSP70, and autophagy may be an effective combination [359].

While targeting HSP70 and autophagy may elevate UPR signaling to a point where it is untenable, an alternative way to also achieve this is by reducing basal UPR signaling. The recent development of small molecule inhibitors of UPR mediators offer the potential to selectively target and block these pathways. Inhibition of IRE1 RNase activity has been shown to exert beneficial effects as either a standalone treatment or in combination with chemotherapeutics in pre-clinical models of triple negative breast cancer (TNBC) and prostate cancer [360,361,362]. The status of basal UPR signaling pathways and outcome of UPR inhibition in RMS have not been extensively studied. However, McCarthy and colleagues recently reported constitutive activation of UPR mediators IRE1 and PERK in a panel of RMS cell lines encompassing both ARMS and ERMS subtypes (McCarthy N et al., 2020). Selective inhibition of IRE1 or PERK resulted in divergent outcomes with ARMS cells displaying a marked reduction in cell proliferation and long-term survival to IRE1 inhibition, whereas ERMS cell lines were more responsive to PERK inhibitors [356]. Further analysis demonstrated reduction in the cell proliferation, which was the consequence of cells transitioning into a non-proliferative senescent state [337]. Whether combination with senolytics is sufficient to trigger death of IRE1/PERK inhibitor-treated RMS cells is an interesting question to address in the future. The ever-expanding literature supports roles for UPR mediators in pro-tumorigenic processes distinct from the maintenance of cell viability. For example, IRE1 signaling has been linked to tumor metabolism, epithelial to mesenchymal transition (EMT), angiogenesis, and the development of chemoresistance [355]. Whether IRE1 or other UPR mediators also impact these processes in RMS remains unanswered.

Treatment for RMS with a combination of vincristine, actinomycin-D, and cyclophosphamide (i.e., VAC, see above for more details) is the favorable chemotherapy approach [40]. Although recent pre-clinical models and clinical trials assessing the efficacy of alternate chemotherapies associated with less toxic side effects, such as temozolomide (TMZ), have shown encouraging results [363] and while chemotherapeutics can be effective in inducing the death of RMS cells, relapse is a significant challenge. Alterations in proteostatic mechanisms such as autophagy or UPR are known to contribute to chemoresistance in many cancers. Induction of autophagy in response to TMZ has been reported in RH30 ARMs cells [183]. Combination with autophagy inhibitors increased TMZ-induced cell death in RH30 ARMs cells, suggesting the benefit of combining chemotherapeutics with autophagy inhibitors [183].

Current findings suggest that proteostatic pathways such as the UPR contribute to the progression of multiple cancers in diverse ways. As of yet, few studies have focused on UPR activation within the setting of RMS. Those that have suggested heightened, constitutive UPR signaling in RMS, but the functional consequences are on the whole unknown. Future studies examining the impact of the UPR on RMS progression and responsiveness to chemotherapeutics are required.

## 9. RMS In Vivo Models

### 9.1. RMS In Vivo Mouse Models

In general, there are four main groups of mouse models used for RMS studies (Figure 5) including: (1) cell-line-derived xenografts (CDXs); (2) patient-derived xenografts (PDXs); (3) environmental-induced mouse models (EIMMs); and (4) genetically engineered mouse models (GEMMs). CDXs are xenograft models in which specific cell lines are subcutaneously injected in immunocompromised mouse models to produce models that are the same as human tumor origin and are classified into two forms of orthotopic and heterotopic models [364]. These are used to simulate human cancer tissue and are commonly used in pediatric RMS research [365]. In PDXs, primary tumor tissue is injected subcutaneously in immunocompromised mouse models to obtain either the cells or tissue pieces (orthotopic or heterotopic) [366]. In GEMMs, specific genetic information (i.e., typically from an oncogene or tumor suppressor gene) is used to produce the model [367]. According to the investigation purposes, different types of germlines or somatic mutations are used to produce different types of GEMMs [368]. In EIMMs, animals are exposed to the mutagens (like oxidative stress, aging, or DNA methylation) to mimic the disease that confers different conclusions about the progress of the mutagenesis [365].

Each of the mentioned animal models is suitable for evaluating the specific types of therapeutic approaches (Figure 6). The CDX model is the commonly used mouse model for investigating the drug mechanism of action; however, these models may fail to recapitulate the disease phenotype. PDX models are adopted mostly for the translational sarcoma research including biomarker investigations, local therapies, targeted therapies, combinatorial chemotherapies, and radiotherapies [369]. To investigate specific research questions, mostly related to carcinogens, EIMM modeling could be used together with CDX mouse models [370]. GEMM models are mostly used in tumor microenvironment screening, tumorigenesis, tumor maintenance, and some other applications related to diagnostics and preclinical testing [368]. In the case of adolescent cancers, utilizing EIMMs is the most powerful approach than other types, while PDX and GEMM models are applied as complementary techniques [368]. Table 6 and Table 7 represent different types of the animal model approaches with their pros and cons.

#### 9.1.1. Genetically Engineered Mouse Models (GEMMs)

There are four different strategies for the generation of GEMMs models (Figure 7) including: (a) spontaneous mutations by targeting the related gene [322], (b) chemical/radiation-induced mutation via utilizing some external modulators [378], (c) retroviral transduction [379], and (d) DNA microinjection.

In ERMS or ARMS, several genetic aberrations occur in specific and nonspecific nucleotide regions [379]. Due to the numerous variations in the RMS-related gene map, genetically engineered mouse models could be ideal candidates for the in vivo study of this disease [379]. Among different mutations, the P53 pathway is one of the most common mutations used for mimicking the cancer case, due to the significant role of p53 from nuclear signaling to apoptosis [365,369]. Different types of models could be generated in this context including the inactivated or mutated p53 models [364], models with overexpression of p53 negative regulators like murine double minute 2 (MDM2) [364,380], and p53 null mice model (that could trigger the Pax3-FKHR chimeric factor to generate ARMS) [381,382].

Additionally, the Ras/Erk pathway, which has a close relation with p53, is also used for ERMS determination. Another study demonstrated the generation of an animal model with loss of p53 through the sonic hedgehog (SHH) pathway. By this, they demonstrated the importance of local injection of mutagenic agents in the generation of different mutagenic patterns, especially in breeding GEMM models [368].

In the generation of GEMM mouse models, the genetic background of RMS should also be well known [368].

GEMM mouse models were used for the assessment of the therapeutic applications of RMS [383]. For instance, NODscid mice, a type of GEMM mouse models known with a severe combined immune deficiency spontaneous mutation, were used for the evaluation of RMS treatment with vincristine (VCR)-loaded liposomes [383]. The results of this study showed prolonged circulation of the nanoformulation in blood and improved tumor accumulation into the targeted site that led to high therapeutic performances [383].

#### 9.1.2. Environmental-Induced Mouse Models (EIMMs)

EIMMs are generated by inducing natural-based mutation using carcinogenic agents and ionizing radiations [384]. While the childhood sarcoma is generated by the genetic variations, the adulthood sarcoma EIMM types are produced via utilizing harsh conditions (to mimic the environmental impacts) [369].

Heavy metals are the most common agents used for studying the toxic effect of environmental pollutants from the mutagenic and carcinogenic perspective. For instance, Gilman et al. confirmed that the intramuscular injection of cobalt and nickel to the rat animal models could lead to RMS development [368,369]. Pyrrolizidine alkaloids are the other agents that are recognized to contribute to RMS formation [364]. The metabolite of these agents is dehydroretronecine, which is a type of natural toxin [385]. Benzenediazonium sulphate (BD) is another carcinogen agent that induces RMS formation in several mouse models. Swiss mice with subcutaneous injection of BD indicated the formation of RMS, fibrosarcomas, and osteosarcomas [385].

The effects of polycyclic aromatic hydrocarbons on RMS development, which result from cigarette smoke, urban air, pollution, or other additional external causes, could be monitored via utilizing Sprague-Dawley rats [381,384]. The CD-1 mice models are used to understand the ionizing radiation effects on RMS formation [378]. β-radiation-exposed mice models demonstrated p53 mutations that lead to RMS development [378]. Combinatorial studies are also performed to understand different co-locations and the development patterns of RMS [386]. For instance, to understand the relation between the immune-dependency and tumor progression, immunocompetent mice models are used in which EIMMs models are used to determine the effects of external factors and different radiation types in tumor growth progression [387]. Despite a lot of research being done, there still remains not enough information about screening the effects of the environmentally related parameters on RMS.

#### 9.1.3. Cell-Line-Derived Xenograft Mouse Models (CDXs)

In comparison with other animal models, CDX mice are low cost and highly available models (near 70% among other animal models), which are feasible for in vitro tests [368]. However, only very aggressive types of RMS could grow in in vitro conditions, and the adaptation of these cells with prolonged viability is required for the tumor stroma reflection. Thus, other types of animal models are generally used as complementary approach with CDX, especially for the therapeutic predictions [379].

In general, excess amounts of fetal calf serum (FCS) are used for the in vitro cancer cell culturing for CDX mouse models [365]. Some of the common cell-line-derived mouse models used for the CDX studies are Rh30 (alveolar rhabdomyosarcoma—ARMS), A204 (embryonal rhabdomyosarcoma—ERMS), HS-SY-II (SySa), TC71 (Ewing sarcoma—EwS), and KHOS (osteosarcoma—OS) [388]. Although all of these cancer cells are not well-adapted in 2D cell culture conditions, they could decrease the activity in in vivo tests [368].

In a study, CDX and PDX models were used to indicate the therapeutical safety of antibody against the B7-H3 receptor to treat solid tumor malignancies including RMS as a pediatric cancer type [366]. Kendsersky et al. demonstrated that the CDX and PDX models should be used together for screening the therapeutical applications, and using one type of animal models could not be enough for the investigation of such therapeutics effects [380].

#### 9.1.4. Patient-Derived Xenograft Mouse Models (PDXs)

In patient-derived animal models, tumors are extracted from the patients and directly inserted into the immunodeficient humanized mice. These models have various advantages, aiding in the deeper understanding of actual cancer biology due to their ability to simulate the natural cancer progression. Mostly studied PDX models for RMS are related to musculoskeletal malignancies. For instance, Igarashi et al. generated successful animal models as orthotopic PDX models for the RMS study. However, collecting the samples and transplanting them into mice models are challenging [368,378,389]. Due to the rarity of some subtypes of RMS, logistic challenging to access these samples is the most problematic case for this modeling approach [367].

Some models are generated and published as in the repositories or some data banks like https://www.europdx.eu/ (accessed on 20 October 2023). These models obtained from the patient-derived tissues or primary cells inclusion into mouse models continue to differentiate to online platforms for the academic platforms and research organizations [367]. Despite the impressive success of the PDX animal models, necessary carefulness should be considered during the research, due to the difficulties to have the actual patient samples and also their limited number, especially for the rare diseases like RMS [370].

Lu et al. studied patient-derived xenograft models in musculoskeletal malignancies by generating PDX mice models. In their study, the main aim was to struggle with the appropriate animal model usage due to the complexity and heterogeneity of musculoskeletal malignancies. The models were generated after receiving the samples from the patient as fresh tumor tissues and were preserved in fetal bovine serum (FBS) and transplanted through mice within 2 h for the highest yield. Soft tissue sarcoma was analyzed using PDX models and reliable results obtained in tumor growth with stable genomic alterations [390]. PDX model application has more successful outputs in the soft tissue sarcoma due to transplantation acceptance of the animals [364]. The stable gene alterations that exist in PDX models lead to determine responsible genetic paths with a deeper understanding such as preserved genetic variations both in bone and soft tissue [390].

### 9.2. Zebrafish Models to Study RMS

In addition to mouse in vivo models, zebrafish models have been employed to study the development, histology, pathogenesis, tumor progression, metastasis, and drug screening of RMS [391]. Since ERMS is the most dominant type of RMS in humans, and it was revealed that fusion gene-negative ARMS are similar (both biologically and clinically) to ERMS, the initial models in zebrafish were mostly designed as ERMS subtype. ERMS is characterized by the mutations in the genes for RAS GTPases, MYOD1, and FGFR4. In fusion-positive RMS, the overexpression of fusion protein PAX3/PAX7-FOXO1, caused by the chromosomal translocation, leads to more aggressive type of RMS or 85% of ARMS [392,393,394]. These two RMS models can be recreated in the zebrafish either with genetic modifications or tumor cell transplantation [391].

To study RMS in vivo, zebrafish hold great advantages compared to other animal models. They are small and can produce numerous offspring from a single breeding, consuming less effort and are economically cheaper compared to the commonly used murine species [395]. A highly attractive trait is their ability to generate tumors with similar histological and genetic features to humans [396]. They also provide the opportunity for high-throughput drug screening by submersing larvae in a bath solution of the drug of choice, as well as transplanting primary tumors (obtained from the patient) into immunocompromised lines [392]. The translucent appearance of zebrafish provides the ability of imaging the tumor growth, shape, size and renewal, and relapse over early larval stages via live in vivo confocal imaging in combination with the fluorescent reporter lines. Moreover, the use of zebrafish mutant lines lacking pigmentation altogether, such as the Casper line [397,398], allow highly tractable observations of tumor growth over larval to senescent stages of zebrafish development. This presents unique opportunities for tracking molecular markers, histogenesis and metastasis of cancer cells, as well as drug screening [376,392]. In particular, zebrafish ERMS has shown histological and molecular mechanisms similar to the human ERMS. This model uncovered the pathways that regulate the RMS growth, its propagation, and self-renewal. Furthermore, transgenic zebrafish can be generated with the help of specific promoters added to the muscle for expressing the fluorescent proteins, which can be conditionally activated uniquely at different stages of muscle maturation, thus marking the state of the muscle cells [374].

Despite clear benefits of the zebrafish model, the zebrafish has several drawbacks. When the RMS model is generated via tumor transplantation, limited cell numbers are transferred to the larval fish, which are grown at 28 °C. Furthermore, xenograft fish models are subjected to the drug screening experiments before the elimination of these cells via the immune system (usually around 10 days of their life), which prevents the visualization of drugs and cellular events during the tumor propagation and metastasis during the long term of experiments [399]. Thus, it seems that zebrafish models are most beneficial for the short-term treatment studies but may suffer for the longer-term experiments. In other words, their response against the therapeutic components could be stated when fishes are in the drug-container dish that makes it hard to estimate the drug uptake route. In this case, adult zebrafish, due to their unique features like low cost, optical clarity, good fecundity, and capability of performing high drug throughput and tumor progression studies, could be ideal candidates to be used as cell-transplantation models [397].

Currently, immunodeficient zebrafish models have been developed to eliminate the mentioned disadvantages of zebrafish for the xenograft RMS studies [397]. Based on this, in the following Sections, we will describe different methods used for the generation of RMS model in the zebrafish, including the mosaic transgenic approach, heat-shock inducible Cre-Lox and Tol2 mediated gene trap systems (Figure 8), and the immune compromised models for the tumor transplantation.

#### 9.2.1. Mosaic Transgenic Approach

In this method, one or more genes were injected into zebrafish embryo at its one-cell stage, to initiate the RMS tumor development in several parts of the fish. The most studied approach to generate ERMS in the zebrafish is to express constitutively active Kirsten rat sarcoma viral oncogene homolog (KRAS) (mutated version called KRASG12D) gene using recombination activating 2 (rag2) promoters [392]. The rag2 is expressed in the progenitors of B and T cells, satellite muscle cells, and myoblasts, but not in the multi-nucleated muscle fibers [391]. Once rag2-KRASG12D and rag2-green fluorescent protein (GFP) constructs were linearized and co-injected into the embryos in the one-cell stage, the expression of KRASG12D was validated with GFP imaging at 10 days of post-fertilization, forming KRAS-driven ERMS [391].

For RMS, labeling of more than one cell type and tumor niche were also achieved. Transgenic zebrafish expressing myogenic factor 5 (myf5)-GFP, myogenin-H2B-mRFP, and mylpfa-lyn-Cyan were able to show the sub-population of developed ERMS tumors, indicating GFP fluorescence for tumor propagating cells (TPCs) and cyan fluorescence for differentiated cells. With this approach, tumor subpopulation and heterogeneity were labeled based on the activation of different promoters (myf5, myogenin-H2B, and mylpfa) at different stages in the muscle development [403]. In another case, the expression of different promoters in the muscle development was achieved with the co-injection of rag2-KRASG12D: myogenin-H2B-RFP: mylz2-lyn-Cyan into mutant fish expressing myf5-GFP at the one-cell stage (Figure 9A). The heterogeneity in the tumor was labeled in the ERMS cells that differentiated early, mid, and late phases at 16 days of their life (Figure 9B,C). After the serial cell engraftment to the syngeneic fish, only myf5-GFP-positive cells were the transferred ERMS to the following recipient (Figure 9D). ERMS tumors hold similar histology in the primary tumor and the second recipient (Figure 9E–J) [404]. In addition to tumor growth, several other parameters were studied with the zebrafish KRAS-driven RMS tumors, which are outlined in Table 6.

Cadherin 15 (cdh15) promoter, which is expressed in the muscle satellite cells, was also utilized to generate KRAS-driven ERMS in the zebrafish. The cdh15-KRASG12D and mylz2-KRASG12D as well as rag2-KRASG12D were injected to double transgenic zebrafish embryo at the single-cell level. No histological difference was observed between rag2 and cdh15 promoters in KRASG12D expression, showing the potential of early muscle progenitor cells, which consist of mostly undifferentiated myoblast-like cells, to develop ERMS. Injection of the mylz2-KRASG12D into fish resulted in tumor propagation, which was similar to that in the mature skeletal muscle [405].

Once primary tumors were generated in the donor zebrafish successfully, cells can be harvested and transplanted to the syngeneic fish to study the tumor propagation and volume. In this assay, ERMS cells expressing fluorescence protein were collected from donor fish and sorted based on the fluorescence-activated cell sorting (FACS) technique. The sorted tumor cells were then injected to the recipient adult fish, either with intraperitoneal or intramuscular ways, imaged over weeks, and used for different analyses like cellular pathways, tumorigenesis, and drug screening [371].

#### 9.2.2. Stable Transgenesis with Heat-Shock-Inducible Cre-LoxP Approach

The Cre-loxP approach is widely studied for gene manipulation and is composed of Cre recombinase enzyme and a pair of short nucleotide sequence called LoxP [5′-ATAACTTCGTATA-GCATACAT-TATACGAAGTTAT-3′). When flanking LoxP sites are recognized by the Cre recombinase, the enzyme cuts and recombines the LoxP sites, resulting in excision, insertion, or inversion of genes located between two LoxP sites. This strategy was applied in the zebrafish model for introducing KRAS-driven ERMS model. Transgenic fish expressing β-actin-LoxP-EGFP-STOP-LoxP-KRASG12D and heat-shock protein 70 (hsp70) with Cre (called hsp70-Cre) translate enhanced GFP (EFGP) ubiquitously (Figure 10). When Cre was encoded by the heat-shock treatment via hsp70-Cre, it recombines two LoxP sites, removing the EGFP and expressing KRASG12D, resulting in KRAS-driven ERMS tumor in zebrafish. Double transgenic zebrafish were able to express EGFP for 24 h (Figure 10B,C) and 44 days (Figure 10D,E) post-fertilization without heat-shock. Upon heat treatment [37 °C, 1 h), the Cre-mediated excision was made for the expression of KRASG12D, showing the tumor formation (Figure 10F,G). Surprisingly, non-heat-shocked transgenic zebrafish also formed tumors with low frequency, due to the hsp70 activation during the fish growth (Figure 10F–H) [402].

#### 9.2.3. Tol-2-Mediated Gene Trap System

Gene trap technology provided by the Tol2 transposon system was used to generate ARMS in the zebrafish, via expression of PAX3/FOXO1 oncogenic fusion protein, to study the in vivo development and tumorigenesis of ARMS. In this transgenesis system, synthetic transposase messenger RNA (mRNA) and Tol2 transposon, promoter and florescent protein containing transposon plasmid, are co-injected into the fish embryos. The donor plasmid containing Tol2 was cut from the Tol2 sites and inserted into zebrafish genome, creating stable transgenesis [377]. This model was used to study a novel target of PAX3/FOXO1, called HES3 transcription activator. To generate ARMS in zebrafish, human PAX3/FOXO1, linked with additional GFP or mCherry via viral 2A sequence, was added into the genome of the fish via Tol2 transposon-based system (Figure 11A). Fusion-positive tumors expressing PAX3/FOXO1 were observed for up to 19 weeks using florescence imaging and tracking of the morphology. Several promoters were used to express PAX3/FOXO1 in zebrafish; among them, CMV, β-actin, and ubiquitin promoters were able to generate primitive neuroectodermal tumors (PNETs), RMS, and sarcoma, respectively (Figure 11B–D). It was also stated that PAX3-FOXO1 fusion and PAX3 alone exhibited different characteristics on the embryonal development of the fish. HER3 (zebrafish ortholog of HES3) expression was observed on the fish injected with the PAX3-FOXO1 vector. The expression of HES3 was linked with the pro-tumorigenic events in mammalian cells, which are also linked with the tumor progression and reduced survival in patients with RMS tumors; thus, HES3 could be a novel target for RMS treatment [401].

#### 9.2.4. Immunodeficient/Compromised Zebrafish Models to Study RMS

Tumor grafting to adult fish is a very powerful tool to study tumor expansion, propagation, and recurrence. Using adult fish also provides injection of a higher number of cells. However, xenograft transplantation often fails due to the activation of immune response of the fish, resulting in loss of tumor cells. For the allograft transplantation or transferring zebrafish RMS tumor cells from one individual to another one with matching immunity, transgenic lines are required, which are from syngeneic background with more than four generations [406]. In this regard, several methods were developed to eliminate the engrafted cell rejection, for instance, using gamma rays could eliminate the immune rejection for about 20 days, whereby dexamethasone can eliminate it up to 30 days during which a sustained drug dose is required [406].

Transgenic zebrafish with compromised immune systems were generated via reduced B and T cells activity. In this approach, rag2E450fs homozygous AB strain-mutant zebrafish were generated for the ERMS engraftment [406]. The rag2 promoter is expressed in satellite cells and myoblasts, as well as progenitors of T and B cells. To generate such a mutant, gene inactivation and engineered zinc-finger nucleases were used to alter the rag2. When α-actin-RFP expressing zebrafish were used as donors, the immune compromised recipients were able to hold the engrafted cells, even if they were transformed to multinucleated muscle fibers; however, the wild type counters did not achieve fiber formation in the 30-day experiment. When myf5-GFP:mylpfa-mCherry double positive transgenic ERMS tumors were transplanted to the immune compromised recipients intraperitoneally, different cell types as well as tumor propagating cells were observed, which showed similar histological features to the donor zebrafish ERMS [406].

For further analysis of the dynamics and heterogeneity as well as the propagation of ERMS, rag2E450fs homozygous transparent Casper fish were utilized. The Casper zebrafish lack melanocytes and iridophores, making them translucent compared to the AB strain. The ERMS tumors growing in the CG1 type zebrafish were successfully engrafted to the mutant Casper fish, and it was revealed that the histology of the tumor was protected against the primary version. To image the dynamics of the ERMS tumors, KRAS-induced ERMS was generated in triple transgenic zebrafish expressing myf5-GFP, myogenin-H2B-mRFP, and mylpfa-lyn-cyan (Figure 12a). Later, ERMS tumors were transplanted intramuscularly to the 3-month-old Casper fish with flk1-mCherry, rag2E450fs expression (Figure 12b). Thus, tumor propagating cells were illustrated by the GFP florescence, while differentiated cells were labeled with AmCyan (Figure 12c) [403].

The xenograft ERMS transplantation was achieved using the mutant Casper fish with the lack of T, B, and natural killer cells. The immunocompromised Casper zebrafish were generated through inducing deficiency in protein kinase DNA-activated catalytic polypeptide (prkdc) and interlukin-2 receptor gamma a (il2rga) via crossing the mutant prkdcD3612fs/D3612fs and il2rgaY91fs/+ in adult Casper (roya9/a9 and nacrew2/w2 mutants). Homozygous inbreeds prkdcD3612fs/D3612fs and il2rgaY91fs/Y91fs were chosen as prkdc−/−, il2rga−/− fish. When GFP-expressing cancer cells were intraperitoneally injected, zebrafish were able to hold the cells, and tumors were growing up to 4 weeks with a death ratio of less than 15%. When RMS cells were transplanted to the immunodeficient zebrafish and mice, both recipients exhibited similar histological profiles. The mutant fish were also able to hold the patient-derived RMS tumor within 4 weeks at 37 °C (Figure 13A). Translucent prkdc−/−, il2rga−/− Casper fish also provided the tracking of RMS cells at a single-cell level over a week. When a drug cocktail including the combination of temozolomide (TMZ) and olaparib was orally administrated to the EGFP-labeled human-RMS engrafted immunodeficient Casper fish, the tumor size decreased (Figure 13B,C). At the end of the drug treatment for 28 days, animals were sacrificed, and tumors were labeled with histology analysis via hematoxylin and eosin (H and E) (Figure 13C), Ki67 staining for cell proliferation (Figure 13D), and terminal deoxynucleotidyl transferase dUTP nick end labeling (TUNEL) assay for the apoptosis (Figure 13E). According to Figure 13C, combination therapy showed elimination of tumor mass compared to TMZ or olaparib alone. The histological analysis showed the overall declines in cellularity in animals administrated with TMZ or olaparib alone and almost complete loss of cell proliferation after three cycles of combination therapy (Figure 13C–E) [397].

## 10. Tissue Engineering Basics

The engineering and manufacturing of replacement tissue is specific to the tissue engineering field. Recently, tissue engineering has obtained much attention in the field of medicine as an alternative to grafts or transplants. Tissue engineering uses a patient’s own cells to generate a functional tissue or organ [407]. Overall, there are three models for tissue engineering, including acellular scaffolds, scaffold-free cell-only designs, and hybrid cellularized scaffolds. The emergence of 3D printing has made significant progress in the field of tissue engineering, as 3D printing allows the fabricated tissue to include multiple cell types [408,409,410], biomaterials [411,412], and growth factors [413].

Indeed, the application of 3D printing technology offers numerous advantages over traditional manufacturing methods in different parts of the biomedical field. For instance, in the case of tissue engineering, 3D printing employs biocompatible materials like hydrogels and bio-inks to create intricate tissue constructs that replicate native tissue architecture. These constructs could then be used as scaffolds for cell attachment, proliferation, and differentiation and so fabricate functional tissues and organs that could be transplanted inside the body or could be used as a disease model. Furthermore, 3D printing has revolutionized drug delivery systems, allowing precise control over drug carrier characteristics. This technology has wide applications in the pharmaceutical industry, offering benefits like enhanced efficiency, complex drug release profiles, multiple dosing options, cost-effectiveness, and personalized drug delivery. Personalized medicine tailors drug delivery systems to individual patient needs, considering factors such as age, weight, organ function, and disease severity. 3D printing at the point-of-care facilitates the creation of customized drug combinations, speeding up drug development and reducing costs. Additionally, 3D printing advances high-throughput drug testing with 3D bioprinted tissue models, enabling early-stage biomolecule screening, saving time and resources. The integration of 3D cell culture models with various technologies allows for a more accurate representation of human biology, revolutionizing the understanding of cellular and molecular pathways underlying human diseases and offering innovative solutions to complex healthcare challenges. By incorporating in vitro 3D cell culture models alongside cell lines, significant advancements have emerged, including the development of innovative technologies such as microfluidic devices, tissue-on-a-chip, and organ-on-a-chip systems. These innovations have reduced the limitations and drawbacks related to the application of 2D cell culture and animal models [414,415,416,417].

The performance of this new technique was enhanced by introducing the 4D printing method, an emerging and cutting-edge technology in which the dimension of time is incorporated into the printed objects. This technique leads to producing materials with the ability of changing their shape or structure during the implantation that could promote seamless integration with adjacent tissue and foster optimal cell proliferation and differentiation. In addition, they present an avenue for the creation of adaptive drug delivery systems designed to release therapeutic payloads in response to precise physiological triggers, such as variations in pH or temperature, ensuring targeted and controlled drug dispersion [418].

### Application of 3D Printing in Muscles and Rhabdomyosarcoma Tissue Engineering and Treatment

In recent years, additive manufacturing (AM) or 3D printing has been widely used in different fields such as aerospace, automobile, construction, and medical science. In the medical setting, 3D printing can be used for three different purposes, including tissue engineering, implants, and surgical planning prototypes. A number of recent studies have highlighted the potential of 3D printing applications in medical tissue engineering [419,420,421], implants [422,423], and surgical planning prototypes [424,425] for the muscles and RMS modeling and treatment.

As a gold standard, surgery and resection of the tumor is the most reliable clinical treatment for RMS. Skeletal muscle has a robust ability to regenerate and remodel following injury largely due to the presence of muscle progenitor cells called satellite cells. However, if the resected damage to muscle is larger than the capacity of regeneration or the size of the primary tumor is particularly large, the structural and functional deficits occur as fibrosis and sometimes fibro–fatty tissue infiltrate muscle tissue with the expansion of resident fibro–adipogenic progenitors (FAPs). Thus, there are times when it is necessary to assist muscle regeneration with adequate functionality and structure at the tumor site. Further, 3D printed scaffolds could help to create platforms with complex microstructures to guide cell alignment and fusion and consequently to regenerate replacement tissue for resected tissue [419,420,421]. Moreover, the application of either natural or engineered biomaterials in the printed scaffolds not only could increase tissue formation but also could improve the functionality of the regenerated tissue [426,427,428,429]. For example, Kim and Kim [428] used a collagen-based bio-ink for skeletal muscle tissue regeneration. They produced 3D scaffolds made of C2C12 myoblast-laden bio-ink using extrusion printing with uniaxially aligned topographical cues. In vitro analysis revealed a high degree of cell alignment and efficient differentiation. Indeed, the presence of collagen in the structure of this scaffold induced the production of some biochemical cues that led to the attachment and growth of cells from one side and aligned the physically designed topography of the cells, from the other side. One of the main targets of fabricating 3D printed scaffolds is creating a microenvironment with the most similarity to the native extracellular matrices. For instance, Kang et al. [430] developed an electroconductive C2C12-laden bioactive bio-ink composed of a phenol-rich gelatin (GHPA) and graphene oxide (GO) for the production of a 3D printed scaffold. In vitro analysis showed that the myogenic differentiation of C2C12 myoblasts was spontaneously facilitated without the inclusion of myogenic differentiation-inducing factors. It was a stable scaffold with microporous structure with proteolytic degradable property that showed long-term culturing ability. The microporous structure of the scaffold and the protein adsorption capability of GO provide the capability of cell attachment, proliferation, and normal metabolism. In addition, GO could induce myogenesis due to its interesting features such as its electrical conductivity, roughness, and the surface oxygen contents. Scaffolds could also be prepared using biomimetic methods that have the ability of replicating the native tissue microenvironment in terms of parallel-aligned structures and the incorporation of biophysical signals. Bilge et al. [431] used this feature in their work to fabricate a 3D printed electroactive scaffold the skeletal muscle tissue engineering. The scaffolds were made of poly(ɛ-caprolactone) (PCL) combined with carbonaceous material (CM), which incorporated the electrical conductivity. This group seeded the printed scaffolds with C2C12 myoblasts and subjected it to an electrical stimulation during an in vitro test. Their results confirmed enhancement in myotube formation in electroactive scaffolds compared to non-conductive scaffolds. It was also observed that the myotube formation and myotube maturity were significantly increased in the CM group following electrical stimulation (Figure 14) [431].

Although 3D printed scaffolds are useful in tissue engineering, they do have some limitations. Generally, implantation of the printed scaffolds, especially hydrogel-based ones, is a difficult process because they are not suturable and do not adhere properly to the host tissues [432,433]. Moreover, for the complex-shaped injuries, it is possible that the printed scaffold will not match the defect site exactly, and consequently, a gap or overlap with the surrounding tissue will occur. Moreover, 3D printing of the scaffolds requires pre-processing operations, such as taking images or impressions of the injured area, creating a 3D model from the 2D images, and generating instructions for printing the desired shape, which are normally time-consuming processes. In addition, printing time varies based on the employed method, material, and complexity of the printed shape. In urgent cases, such as those resulting from traumatic injury, the initial surgical intervention should be done quickly after the accident [432]. Considering the time needed for preparing the scaffold, secondary surgery would be required to implant the scaffold into the injured area. To overcome this problem, some researchers have tried to develop a mobile bioprinter with in situ printing ability [432,434,435,436]. Russel et al. [432] proposed a mobile extrusion-based bioprinter for in situ printing in the case of volumetric muscle loss (VML) (Figure 15).

This group employed a gelatin-based hydrogel, printed it directly into the defect area, and cross-linked in situ. The suitability of the used materials was confirmed by several in vitro and in vivo tests. The results of hematoxylin and eosin (H and E) staining of the harvested samples from murine models with VML injuries showed adequate adhesion of the printed gel to the surrounding tissues. Moreover, there were no signs of rupturing in the structure of the gels, which confirmed its suitable mechanical properties (Figure 16) [432].

Another application of 3D printing in medical science is the production of patient-specific implants, and there are several reports of this approach used in the following treatments of RMS [422,423]. For instance, O’Sullivan et al. [423] produced a customized 3D printed eye cover for an 18-year-old man with left maxillary ARMS utilizing a biocompatible material. They utilized a 3D scanner to map the surface of the patient and then created a 3D model of the implant. With the advantages of AM, the entire process from beginning to the end of printing was less than 72 h. Thus, AM can assist to create custom-built implants for the patients in palliative care to meet rare and difficult clinical challenges [423].

As noted above, the main clinical treatment for RMS is tumor resection via surgery. Complete tumor resection is normally difficult to achieve and any error in the resection surgery may cause damage to the neighboring tissues and long-term sequelae. Thus, development of surgical skill is imperative. To this end, either physical models or surgical planning prototypes (i.e., phantoms), can be used. In the recent years, several studies have been done on the manufacturing phantoms using AM [424,437,438,439,440,441]. Among the AM methods that have been used in different studies, material jetting, stereolithography (SLA), and fused filament fabrication (FFF) are the most common methods used for producing phantoms [442]. The advantages and disadvantages of these methods are presented in Table 8.

The additively manufactured phantoms could be used for two different goals: visualization of the soft tissues and mimicking the modeled tissue for its shape and mechanical properties. In the first goal, the phantoms are used to give some insight about the tissue geometry to the surgeons before the operation. In this case, mechanical properties of the phantom are not important, and the accuracy of the tissue geometry is the only matter of fact [442]. However, for the second goal, the produced phantom should have both mechanical properties and geometrical parameters the same as the targeted tissue [424,442]. The FFF method is a cost-effective type of additive manufacturing, which is suitable for producing phantoms used for visualization applications, since the common material used in this method are rigid. However, most of the produced phantoms are mono-material and mono-color, which are not suitable to obtain a good insight from different parts of the tissues [424,425].

Recently, some researchers have combined the advantages of the AM with the other manufacturing methods to produce multilateral and multi-color phantoms with mechanical properties similar to the live soft tissues. Tejo-Oreto et al. [425] created a soft surgical planning prototype for a biliary tract RMS. To this end, they used computed tomography (CT) to obtain 2D images from the liver that were then overlapped to reconstruct a 3D model (image segmentation). To distinguish between different anatomical structures, divers parts of the prototypes are highlighted with different colors as follows: (1) red for the hepatic artery, (2) purple for the portal vein, (3) blue for the vena cava, (4) green for the gallbladder, and finally, (5) the brown color corresponds to the tumor (Figure 17A) [425]. According to the 3D models of the tissue that were created from the CT images, the inner parts of tissue and mold were designed to produce the main body of the tissue (Figure 17B). Since the surface quality of the inner parts was important, the inner parts were produced by the SLS, while the molds were produced by FFF. Then, the inner parts were located in their position inside the mold and phantom was produced using the casting method (Figure 17C). They suggested that 6%wt PVA (poly vinyl alcohol)/1%wt PHY (Phytagel)-1FT (freeze–thaw cycles) and 1%wt agarose have the highest similarity to the liver tissue in terms of the mechanical properties. They also have investigated the hardness and mechanical properties of different composites using dynamic mechanical analysis (DMA) tests and Shore hardness tests to obtain the optimum combination for mimicking the liver tissue. The CT result showed that the produced phantom geometry had less than 1% difference with the designed model, and the total cost to produce phantom was relatively lower than that of other technologies [425].

In addition to the two areas that phantoms can help for the skeletal muscle and RMS modeling and treatment, recently, phantoms have been used for patient education as well. Previously, to increase the interaction between the doctors and patients or even their families, CT or MRI images of the tissues or tumors used to be employed. Although it was fruitful, using a 3D prototype will lead to better results in comparison with the 2D pictures and could educate patients about their treatment progress [443,444,445].

Additive manufacturing or 3D printing helped to resolve a lot of limitations such as reproducibility, accuracy, and precision in the manufacturing. However, 3D printed structures despite being native tissues are not active in response to the external stimuli, i.e., dynamic 3D constructs [446,447]. In recent years, application of smart materials (stimuli-responsive material) in 3D printing have led to four-dimensional (4D) printing as a new technology that can produce smart structures that are able to respond to external stimuli that add a new dimension to the 3D printed structures [447]. Application of biocompatible smart material in 4D printing made it possible to use this technology in different fields of biomedical application such as tissue engineering [448,449,450,451,452,453,454,455], implants and medical devices [456,457], and soft robotics [458,459]. Some studies have been done in this field for engineering of different tissues like bone [448], neural conduits [449,450], vascular structures [451,452], and muscle tissue [453,454,455]. Further, 4D printed scaffolds, which can undergo morphological changes in a pre-planned way, could be beneficial for the muscle tissue engineering and RMS treatment [455,460]. For instance, Constante et al. [460] have employed 4D printing to fabricate hollow scroll-like cellular structures with a specific orientation of myoblast. They have combined extrusion printing (methacrylated alginate) and melt electro writing (polycaprolactone) to fabricate the shape-morphing scaffold. The external stimuli in this study changed the Ca^2+^ ion concentration in the surrounding medium since the methacrylated alginate gel has a high sensitivity to this ion. First, they printed the planar scaffold using methacrylated alginate gel by extrusion printing and then added polycaprolactone fibers in the printed scaffold using melt electrowriting to add enough mechanical support to the whole structures of the scaffold. After photo crosslinking of the hydrogel, the C2C12 mouse muscle cells were seeded in aqueous media into the scaffold. Then, by changing the Ca^2+^ ion concentration in the medium, the printed structure starts to fold and form a tubular structure with the enclosed cells. The results of the in vitro study show that using a scrolled bilayer scaffold could help to increase the viability and proliferation of myoblasts cells. Moreover, it is possible to control the cell orientation very well by adding the patterned surface generated by PCL fibers, which is hard, if not impossible to do, on the hydrogel layer without fibers [460].

There have been very limited studies about the application of 4D printing in the field of muscle and RMS modeling and treatment. It should be noted that this technology is still in its early stages. One of the important areas of research in 4D bioprinting is the development of smart bio-inks. Most of the available smart material are triggered by stimuli like changes in temperature and pH that are not suitable for biomedical application [461]. In addition, development of new materials may need to advance the printing methods technologically. In the case of RMS and muscle tissue modeling and treatment, application of 4D printing could help in the development of phantoms where their mechanical and rheological properties can change in response to the external stimuli. This can help the surgeons to obtain a better insight of the tissue or tumor behavior during the resection surgery.

One feature of RMS cells is their invasion and migration into the neighboring and distant tissues. Therefore, understanding the interactions involved in mediating metastasis of RMS cells is of importance. The experiments using 3D cell culture of RMS have shed some light on the cell–cell associations involved in tumor invasion [462]. To introduce tumor heterogeneity, malignant ERMS cells were co-cultured with normal human skeletal muscle myoblast (HSMM) cells using a cell sheet strategy. When the number of ERMS cells is lower compared to HSMM cells, cell sheet disruption occurs. However, sheets containing only ERMS or HSMM cells are intact. Further investigations revealed that malignant cells are able to interfere with HSMM cell alignment. Hence, ERMS cells negatively affect their surrounding tissues and cells due to their aggressive behavior. Furthermore, as muscles are affected by invasive ERMS cells, targeting them can be of importance in inhibiting cancer metastasis (Figure 18) [462].

The development of 3D culture systems also allows us to evaluate the impact of chemotherapy and other therapeutic modalities in the treatment of RMS. Recently, we developed a 3D culture system for ARMS containing thermally cross-linked collagen disc and ARMS cells that have similar biochemical parameters of tumor extracellular matrix (ECM). This method is able to determine the potential of chemotherapeutic agents in ARMS suppression. Furthermore, we can evaluate apoptosis and autophagy induction in ARMS by chemotherapeutic agents in this model (Figure 19) [463].

## 11. Conclusions and Perspectives

Despite advances in RMS therapy, drug resistance and tumor recurrence continue to be long-standing clinical issues. Molecular research into RMS pathology has identified a strong link between genetic and epigenetic alterations and cell growth, proliferation, differentiation, and apoptosis. The identification of PAX-FKHR fusion genes has shifted our research focus with the goal of elucidating pathways that lead not only to the chromosomal translocations, but also on how these oncogenic fusion proteins alter the cell phenotype and can be exploited pharmacologically. This approach could allow for the identification of biomarkers that could be applied to individualize the targeted therapy and improve RMS prognosis.

The integration of new therapeutic agents into the currently recommended treatment regimens seems promising. The aforementioned studies have already led to multiple large-scale clinical trials. The upcoming results will show whether precise targeting of apoptosis can be successfully transferred into a clinical setup. However, numerous agents targeting CDK4–CDK6, MEK, or TRK might have an even larger impact on the treatment efficaciousness. Moreover, the recent discoveries of microRNAs during RMS differentiation along with the role of cell surface receptors preferentially expressed in the RMS cells may enhance personalized therapy through the use of antagomirs or monoclonal antibodies, respectively.

At the present time, there are few efficient therapeutic alternatives available for RMS patients and the only standard treatment protocol for the three subgroups of RMS including low-, intermediate-, and high-risk is the VAC-triple therapy. The VAC regimen consists of an alkylating agent such as cyclophosphamide or ifosfamide along with vincristine and dactinomycin (actinomycin-D) [5,276,277,464]. Considerable research efforts have been made to improve the treatment outcome of pediatric metastatic RMS by adding one or more anticancer compounds to the standard VAC chemotherapy; however, to date, none of the new regimes have been more effective than the VAC protocol [62]. The IRS-IV study tested VAC (vincristine, dactinomycin, and cyclophosphamide) therapy compared to the VAC combined with vincristine, topotecan, and cyclophosphamide (VAC/VTC) in patients with intermediate-risk RMS. There were no significant differences in the effect of VAC against VAC/VTC between the risk groups [6]. Another study investigated VAC therapy compared to vincristine, dactinomycin, and ifosfamide (VAI) and vincristine, ifosfamide, and etoposide (VIE) in patients with intermediate-risk RMS. This study showed that there was no significant difference in the overall 3-year survival rate among patients who received VAI and VIE and those who received only the VAC regimen [5]. In a clinical trial by the International Society of Pediatric Oncology (SIOP) in Europe, 457 patients aged 14 years with high-risk nonmetastatic soft tissue sarcoma were treated with either ifosfamide, vincristine, and dactinomycin (IVA) or IVA plus carboplatin, epirubicin, and etoposide (ICE) for 27 weeks. They reported no survival advantage but toxicity for this treatment protocol [465]. Other studies assessed VAC therapy followed by pre-administration of ifosfamide/etoposide (IE) [466], vincristine/melphalan (VM) [466], and ifosfamide/doxorubicin (ID) [466] in the patients with high-risk RMS. They found that the overall 3-year survival rate with the IE, ID, and VM-containing regimen was noticeably better than VAC regimen alone [466]. The Children’s Oncology Group evaluated irinotecan alone and in combination with vincristine in intermediate-risk patients with RMS. They were unable to document any improvement in the survival rate in RMS patients treated with irinotecan plus vincristine versus irinotecan alone [13]. However, the vincristine, irinotecan, and temozolomide combination has shown synergistic antitumor activity against RMS, which is now the standard treatment protocol for children and adults with relapsed or refractory RMS in Europe [467]. The bottom line is that multidrug resistance (MDR) often occurs after prolonged chemotherapy, which in turn leads to refractory cancer and tumor recurrence. Therefore, proliferation-inhibiting and apoptosis-inducing in MDR tumor cells could be a new weapon for preventing the development of MDR in cancer therapy. Autophagy, a self-degradative process, generally arises during the treatment of multidrug-resistant tumors. In this regard, genetic and pharmacological autophagy inhibitors are used along with therapeutic agents in various malignancies including B cell lymphomas [468], colorectal cancer [469,470], myeloid leukemias [471], ovarian cancer [472], pancreatic cancer [473,474], renal cancer [475], bladder cancer [476,477], cervical carcinoma [478], and lung cancer [479,480,481], which in turn lead to tumor growth impairment and therapeutic sensitivity improvement. On the other hand, as shown in Table 5, as a double-edged sword, autophagy may lead to the death of MDR cancer cells in which apoptosis pathways are inactive. Therefore, more investigations about the combination of autophagy modulators with therapeutic agents are urgently needed in the treatment of various cancer types. To date, numerous studies have been carried out by different combinations of autophagy inhibitors and activators and chemotherapeutic drugs. Nevertheless, studies on the application of this treatment strategy for combatting the development of chemoresistance are very limited for RMS.

The UPR is endogenously upregulated in RMS [356,357]. Thus, targeting UPR could be one of the promising future therapeutic approaches in this rare childhood cancer. As an example, targeting IRE1-sXBP1 or PERK axis of UPR using MKC8866 or PERK inhibitors in combination with apoptosis-inducing chemotherapy may improve the efficiency of chemotherapy in this deadly disease. On the other hand, the UPR is a regulator of both apoptosis and autophagy [341]; therefore, targeting UPR could be a good strategy to potentiate the effect of chemotherapy compounds that affect these pathways.

One of the major challenges in cancer investigations, including RMS, is tumor stiffness during the tumor growth. As tumor cells grow, they remodel their environment by altering the protein content and nearby cell type in their ECM [482]. This feature associated with the changes in microenvironment stiffness results in altering the cellular behaviors. In addition, autophagy is also involved in ECM mechanotransduction, thus it was suggested that increasing autophagy is recorded in normal mammalian cells with increased matrix stiffness [483]. This brings the idea of autophagy involvement in the mechanical regulation of cancer cells. We believe the need for modeling of 3D tumor environments with different ECM stiffness has emerged to mimic tumor environment to test the changes in autophagy and then the chemotherapy response in RMS. Although there are animal models for RMS including mice and zebrafish, it is hard to mimic the exact mechanical environment in animals. It was also stated that measuring the tumor stiffness in the animals are hard, complex, and complicated methods [484]. Unlike the disadvantages and complexities of the animal models, using 3D culturing techniques with hydrogels provides easy and better understanding of the studies related to the cell behavior and drug screening under different mechanical stressors in RMS models.

## Figures and Tables

**Figure 1 cancers-15-05269-f001:**
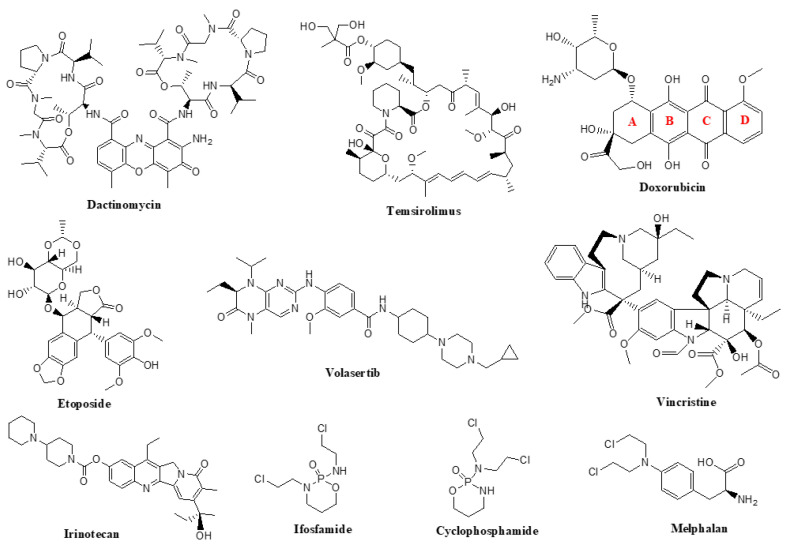
Chemical structures of frequently used chemotrophic agents for the treatment of RMS. In the case of doxorubicin, this planar tricyclic system is called anthracyclinone, composed of aromatic rings B, C and D that can fit between the two DNA strands, orienting itself perpendicular to the long axis of DNA.

**Figure 2 cancers-15-05269-f002:**
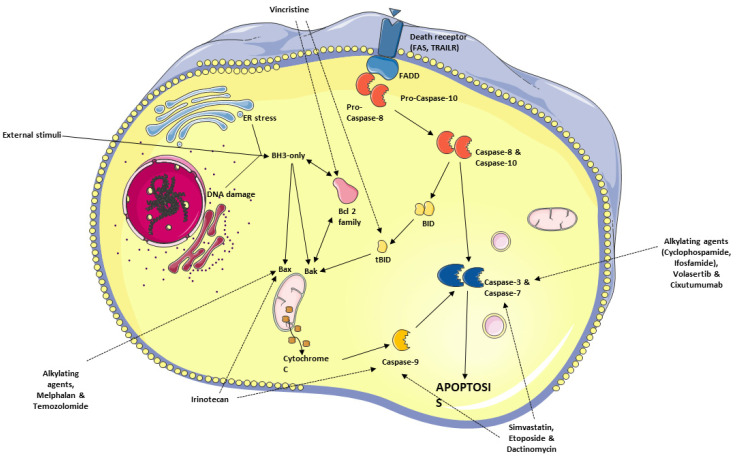
A schematic overview of the molecular mechanism of apoptosis. Apoptotic cell death can be triggered by various stimuli, both intracellular (DNA damage or endoplasmic reticulum stress) and extracellular (TNF-α, TRAIL). The extracellular pathway is initiated by the binding of ligands to the plasma membrane-located death receptors. The stimulation of death receptors induces caspase 8 and caspase 10 activation with subsequent downstream cleavage of effector caspases, caspase 3 and caspase 7. The intrinsic pathway is regulated by the Bcl-2 family. It consists of pro-survival and pro-apoptotic members, the latter of which belong to BH-3 only proteins. Overexpression of BH-3 only, as well as other pro-apoptotic members of the Bcl-2 family, initiates programmed cell death. Bax and Bak are the main effectors of the Bcl-2 regulated pathway. When activated, they increase mitochondrial membrane permeability and allow for the release of apoptogenic cytochrome *c* into the cytosol. Cytochrome *c* release prompts the formation of protein complex called apoptosome, which turns pro-caspase 9 into caspase 9. Then, caspase 9 activates effector caspases, leading to apoptosis. Dotted arrows represent the interactions between chemotherapeutic agents and respective proteins involved in the process of apoptosis. Abbreviations: alkylating agents—cyclophosphamide and ifosfamide, BH3-only—Bcl-2 homology 3 only, BID—BH3-interacting domain death agonist, ER—endoplasmic reticulum, FADD—Fas-associated protein with death domain, tBID—truncated BID, TRAILR—TNF-related apoptosis-inducing ligand receptor.

**Figure 3 cancers-15-05269-f003:**
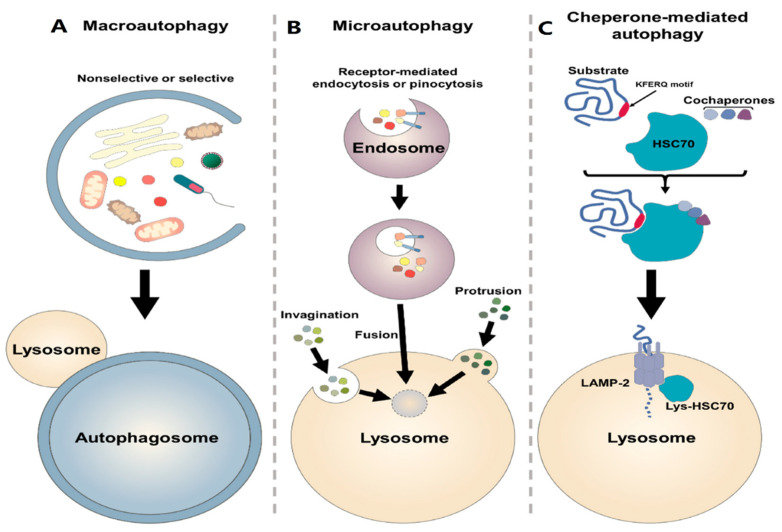
Autophagy is divided into three major types. (**A**) Macroautophagy, the lysosome is fused into the autophagosome to digest substrates. (**B**) Chaperone-mediated autophagy (CMP), HSPA8 complex detects KFERQ motif on the substrate proteins and transports them to the lysosome. (**C**) Microautophagy, the substrates are directly transported to the lysosome.

**Figure 4 cancers-15-05269-f004:**
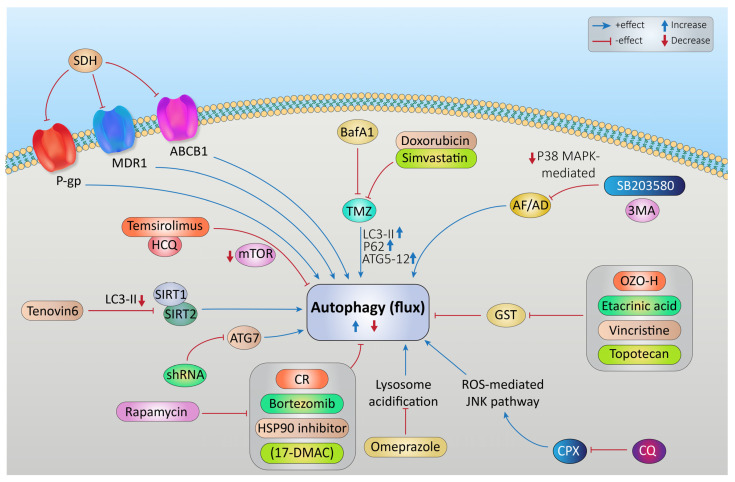
A schematic overview of autophagy targeting in relation to RMS. Different therapeutic strategies produce synergistic or additive effects to treat RMS cancer cells and enhance their response to anticancer compounds. The treatment regimens can affect the autophagy flux. For example, TMZ, AF/AD, ATG7, Ros-mediated JNK pathway, lysosome acidification, SIRT, and ABCC subfamily (P-gp/MDR1/ABCB1) have increasing effect on autophagy flux; but inhibitory treatments like doxorubicin, simvastatin, vincristine, omeprazole, bortezomib, tenovin, and tesirolimus revert the autophagy process by affecting the targets mentioned above.

**Figure 5 cancers-15-05269-f005:**
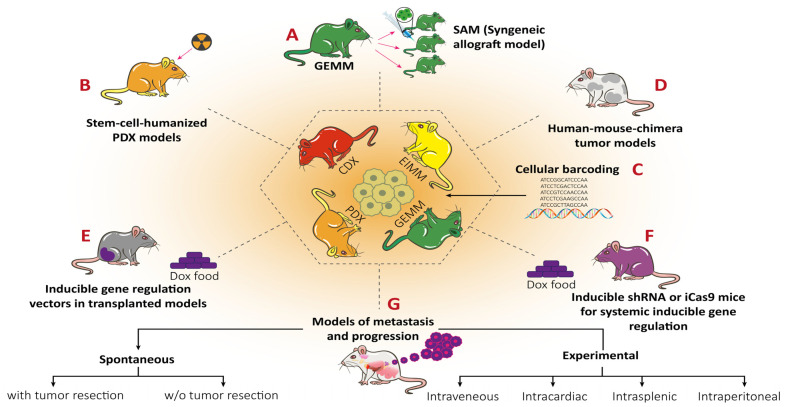
Descriptive RMS animal model chart. (**A**) Genetically engineered mouse models (GEMMs) demonstrated as one modeling approach used in RMS and the production of the model indicated as the most common technique named syngeneic allograft model (SAM) to monitor how tumor cells behave in the presence of immune response. (**B**) Stem-cell humanized patient-derived xenograft mouse models (PDXs) are commonly used as combinatorial approach with cell line-derived xenograft models (CDXs) as complementary study. (**C**) Cellular barcoding is demonstrated as one of the most important steps in producing targeted mouse models especially in GEMM and environmentally induced mouse models (EIMMs). (**D**) Human–mouse chimera is a type of EIMM-dependent mouse model, which could be used for RMS studies. (**E**) Transplantation of tumor tissue into the models by inserting gene regulation vectors for producing PDX models. (**F**) Inducible mouse model generated via gene regulation by iCas9 and shRNA. (**G**) Different types of divergent methods used for producing metastatic and progressive mouse models. Reprinted from [368] with permission from MDPI.

**Figure 6 cancers-15-05269-f006:**
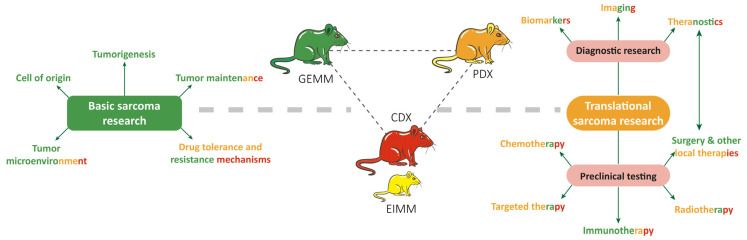
Different applications for sarcoma mouse models. All colors represent the most proper animal modeling approach. GEMM is presented as green, PDX is presented as orange, CDX is shown as red, and EIMM is shown as yellow. EIMM cannot be used by itself, which needs some additional mouse models for the evaluation to be named as precise. All colored letters in green present the certain application field of GEMM, while yellow (EIMM), red (CDX), and orange (PDX) colored letters present the distinct application fields, which are specific to the mouse model. Reprinted from [368] with permission from MDPI.

**Figure 7 cancers-15-05269-f007:**
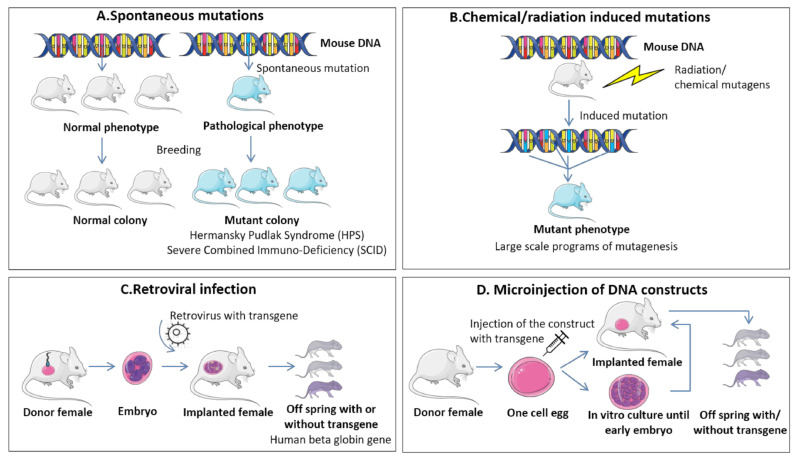
Descriptive chart for GEMM modeling approaches. (**A**) Spontaneous mutations are applied to generate randomized mutant colonies by targeting the gene. (**B**) Externally induced mutation model generated by chemicals or radiations. (**C**) Retroviral transduction is the technique used to breed the transgene via retrovirus. (**D**) Microinjection of DNA constructs to donors. In vitro culture is the way of growing the egg and implanted female generates offspring with transgene. Reprinted from [322] with permission from MDPI.

**Figure 8 cancers-15-05269-f008:**
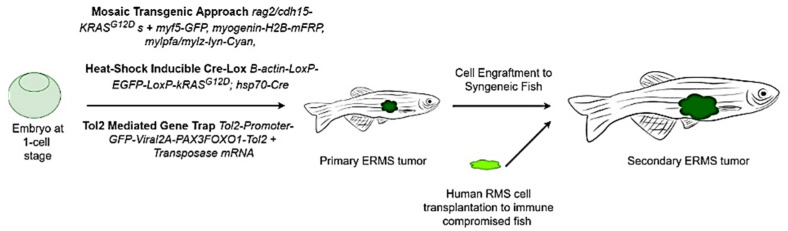
Overview of RMS generation in zebrafish [400,401,402].

**Figure 9 cancers-15-05269-f009:**
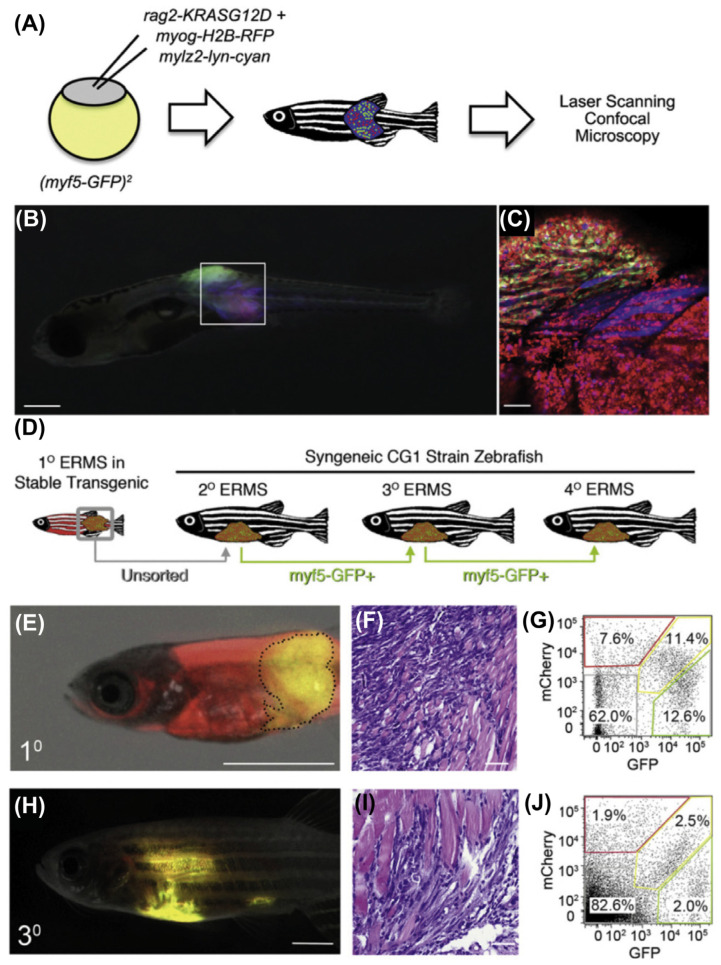
*KRAS*-induced ERMS in zebrafish. (**A**) Co-injection of *rag2-KRAS^G12D^*, *myogenin-H2B-RFP,* and *mylz2-lyn-cyan* into *myf5-GFP* stable transgenic embryos at the one-cell stage. (**B**,**C**) A triple-labeled ERMS at 16 days of life. (**D**) Serial transplantation of *myf5-GF-* positive ERMS propagating cells. (**E**–**G**) A primary ERMS arising in syngeneic *myf5-GFP*; *mylz2-mcherry* transgenic zebrafish at 35 days post-fertilization. (**E**) Fluorescent and bright-field images of transgenic fish, (**F**) tumor histology, (**G**) FACS result of labeled ERMS cells, (**H**) fluorescent and bright-field images of engrafted fish, (**I**) histology of tumor, and (**J**) FACS of isolated ERMS cells. Adapted from [404] with permission from Elsevier.

**Figure 10 cancers-15-05269-f010:**
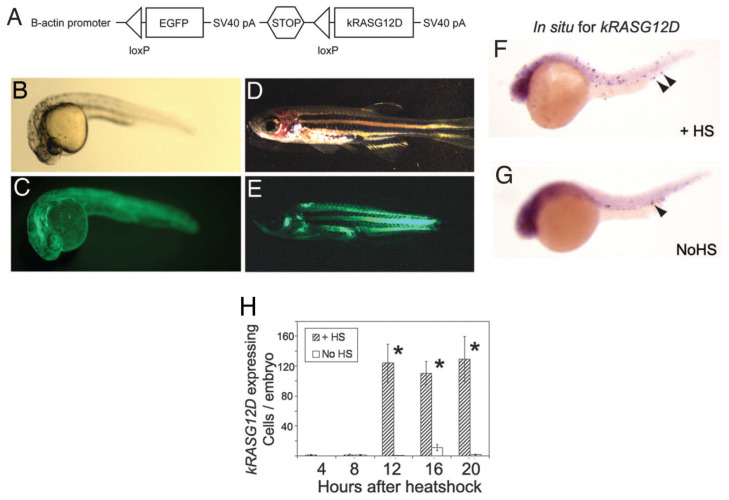
*Cre*-mediated *KRAS^G12D^* transgene expression in zebrafish. (**A**) *β-actin-LoxP-EGFP-LoxP-kRAS^G12D^* transgene illustration. Transgenic zebrafish at 24 h post-fertilization (**B**,**C**) and 44 days post-fertilization (**D**,**E**). (**F**) Heat-shock was performed for *KRAS^G12D^* in situ hybridization of the transgenic embryos (double) at 24 h post-fertilization and (**G**) without heat-shock. *KRAS^G12D^* expressing cells were annotated with arrows. (**H**) The number of cells with *KRAS^G12D^* expression in single embryos with heat treatment from 4 to 5 h post-fertilization and analyzed at 8, 12, 16, 20, and 24 h post-fertilization (* is related to the *p* < 0.001). Abbreviations: +HS = heat shock, NoHS = non-heat shocked. Reprinted from [402] with permission of PNAS.

**Figure 11 cancers-15-05269-f011:**
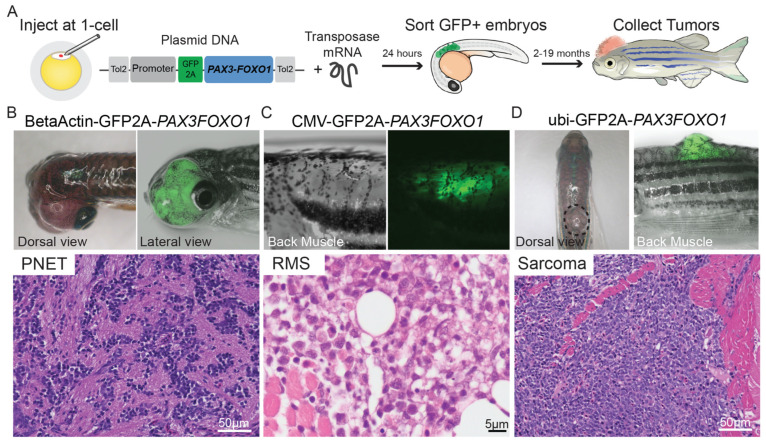
Human PAX3-FOXO1 expression in zebrafish with different promoters using Tol2 trap gene system. (**A**) The *Tol2-promoter-GFP-2A-PAX3-FOXO1-Tol2* was injected to the zebrafish at their single-cell stage, and after evaluating the expression of GFP in 1-day embryos, they were grown up to 19 months. (**B**) Production of PNET in the wild-type genetic zebrafish by PAX3-FOXO1 expressed by *β-actin*. (**C**) Creation of RMS in tp53M214K/M214K-sensitized genetic background via expression of PAX3-FOXO1 by CMV promoter. (**D**) PAX3-FOXO1 via ubiquitin promoter generated a non-differentiated sarcoma in the wild-type genetic zebrafish. Reprinted from [401].

**Figure 12 cancers-15-05269-f012:**
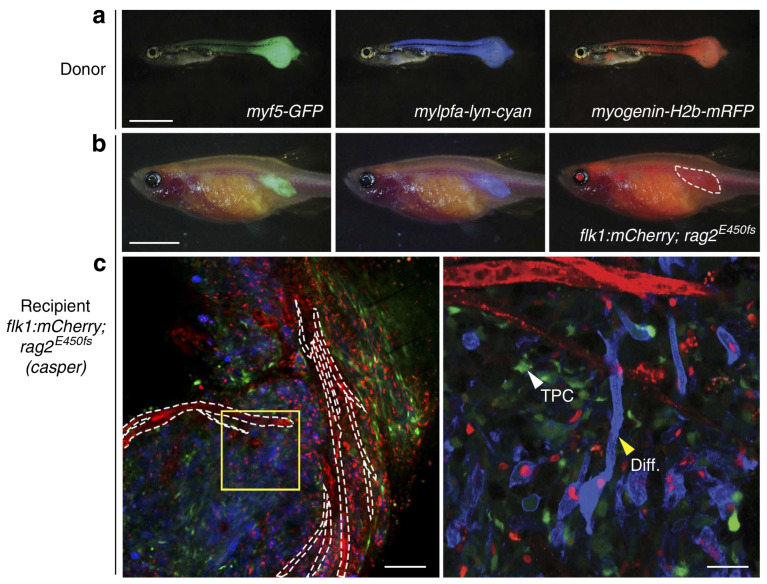
Multichromatic imaging of ERMS tumor heterogeneity at single cell resolution. Florescent images of (**a**) ERMS tumor in different types of zebrafish (*myf5-GFP*, *myogenin-H2B-mRFP*, and *mylpfa-lyn-cyan*) and (**b**) *flk1-mCherry*, *rag2^E450fs^* Casper fish intramuscularly engrafted with fluorescently labeled ERMS after 4 weeks of transplantation (n ¼ 4 animals). (**c**) Confocal images of *mCherry*-labelled vasculature indicated via dashed lines illustrated with X100 magnification (left) and X400 magnification (right). The number of differentiated cells (Diff) was less than the TPCs. *Myosin*-expressing differentiated cells (Diff.). Reprinted from [403].

**Figure 13 cancers-15-05269-f013:**
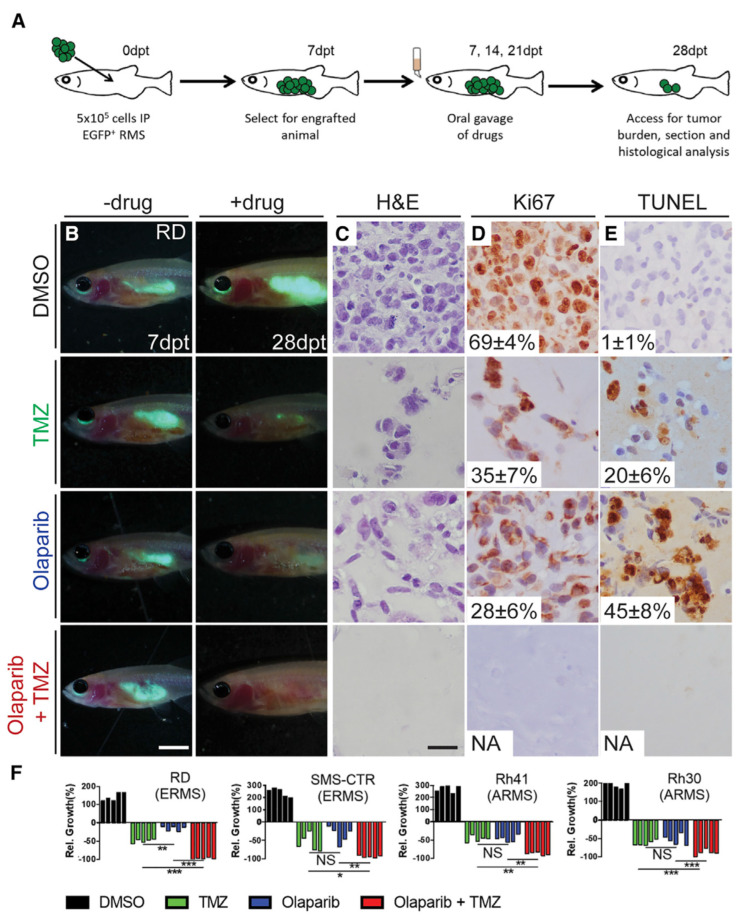
TMZ and olaparib reduce the human RMS tumor growth in immunodeficient zebrafish. (**A**) Experimental design for growing GFP-expressing RMS cells in fish. Fish were selected at 7 days of post-transplantation (dpt), dosed with the drugs at 7, 14, and 21 dpt, sacrificed at 28 dpt. (**B**) Fluorescent images of tumor growth in engrafted animal after 7 days (before drug administration (left)) and 28 days (after 3 times of drug administration (right)) of post-transplantation. Histopathological analysis of RD engrafted sections stained by (**C**) hematoxylin and eosin, (**D**) Ki67, and (**E**) TUNEL. (**F**) Relative growth of ERMS and ARMS cell lines after drug administration. * *p* < 0.05, ** *p* < 0.01, *** *p* < 0.001, Student’s t test. NS: Not significant. The scale bar represents 0.25 cm for (**B**) and 50 mm for (**C**–**E**). Not applicable (NA). Reprinted from [397] with the permission of Cell Press.

**Figure 14 cancers-15-05269-f014:**
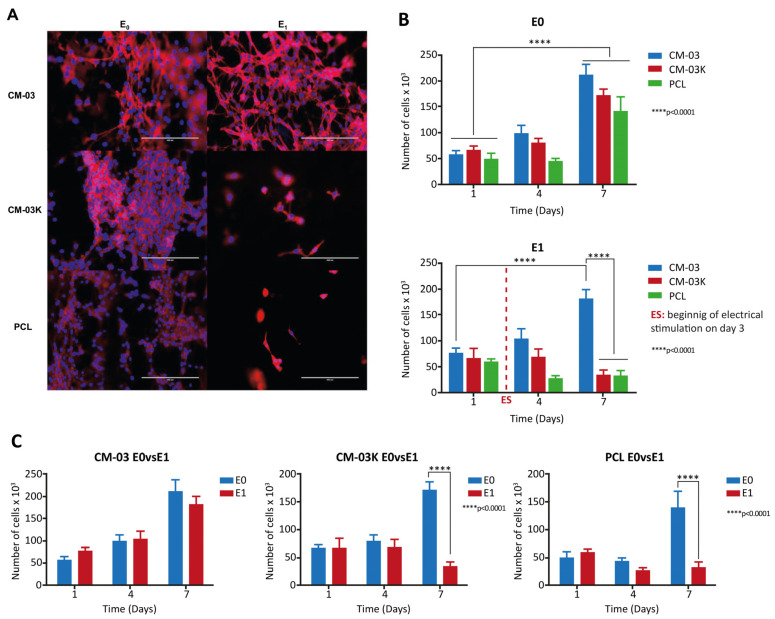
(**A**) Confocal images of cell-seeded scaffolds after 7 days using Phalloidin (red)/DAPI (blue) staining. The results of cell viability test (alamar blue) of (**B**) E0 and E1 groups without/with electrical stimulation (1.5 V), respectively, and (**C**) cells treated with different formulations. Reprinted from [431].

**Figure 15 cancers-15-05269-f015:**
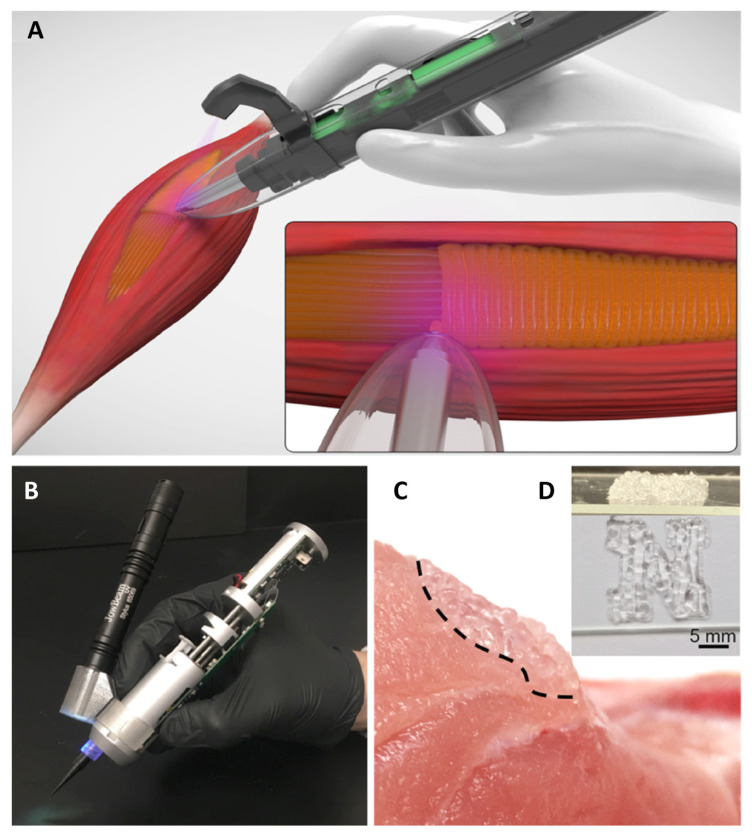
In situ printing of scaffolds using a handheld bioprinter. (**A**) Bioprinting of cell-laden hydrogels for the VML injury treatment. (**B**) A handheld bioprinter, which is able to crosslink the printed hydrogel scaffolds in situ using the provided UV light source. (**C**) Scaffold printed on a non-flat porcine skeletal muscle. (**D**) Printing an N-shaped scaffold on a glass slide. Reprinted from [432] with permission from ACS Publications.

**Figure 16 cancers-15-05269-f016:**
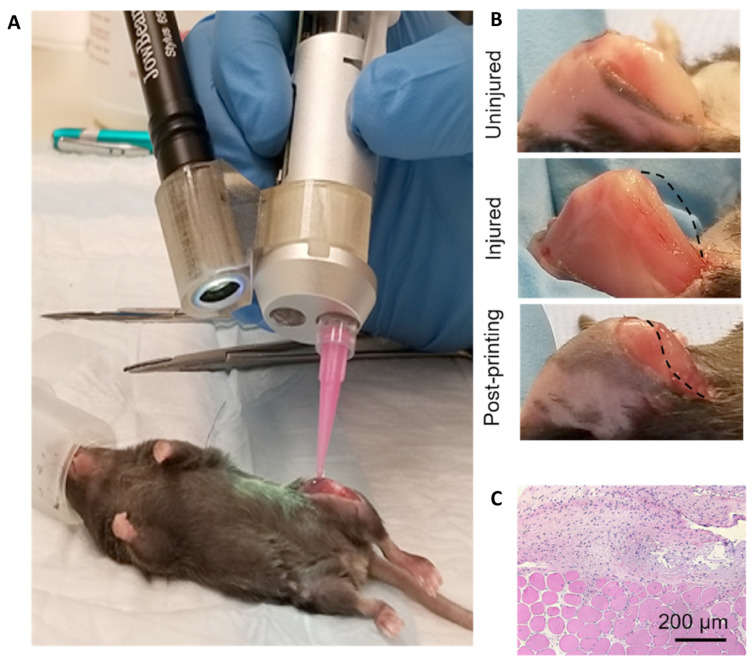
Application of in situ printing in murine model with VML injuries. (**A**) Implanting the GelMA hydrogels into murine VML injury through in situ printing method. (**B**) Before and after VML surgery, and after printing the GelMA hydrogel in the injured site. (**C**) Histopathological analysis of the interface of the printed scaffold interface and the skeletal muscle tissue 4 weeks after surgery. Reprinted from [432] with permission from ACS Publications.

**Figure 17 cancers-15-05269-f017:**
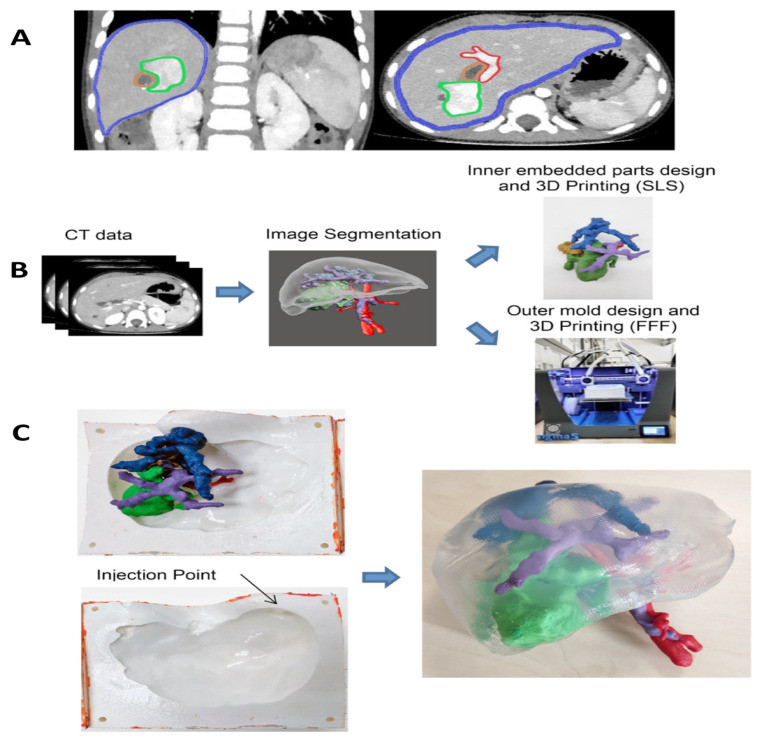
(**A**) Coronal (left) and axial plane (right) of liver tissue CT image; blue color is related to liver, and red and brown colors are related to the portal system and intrahepatic biliary tract tumor, respectively. Green was also used for determining the rest of the tumor. (**B**) Model design procedure. (**C**) After production of all parts, inner parts were located inside the printed molds, and then 3D manufactured surgical planning prototype was produced by casting method using 1%wt agarose. Reprinted from [425] with permission from ACS Publications.

**Figure 18 cancers-15-05269-f018:**
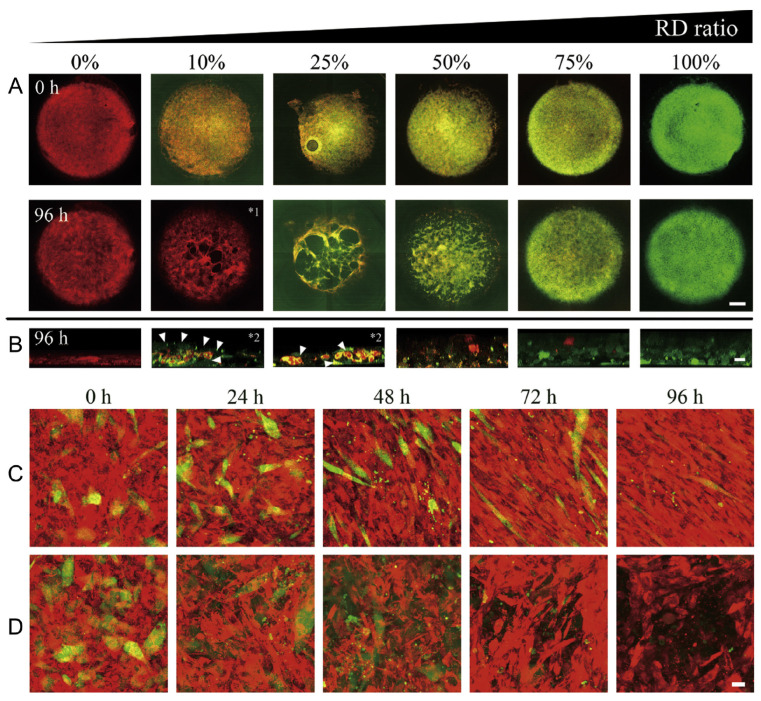
Demonstration of the sheet structure distortion with alterations in the RD ratio: (**A**) sheet morphology time course (t = 0–96 h and scale bar = 2 mm), (**B**) the RD (green) and HSMM (red) localization in heterogeneous sheets (t = 96 h) and the time course of HSMM cell in multilayered cell sheet, and (**C**) the HSMM sheet (both green and red cells are HSMMs) as the control, and (**D**) the mixture of 10% of RDs (green) and HSMMs (red) sheet. Reprinted from [462] with permission from Elsevier.

**Figure 19 cancers-15-05269-f019:**
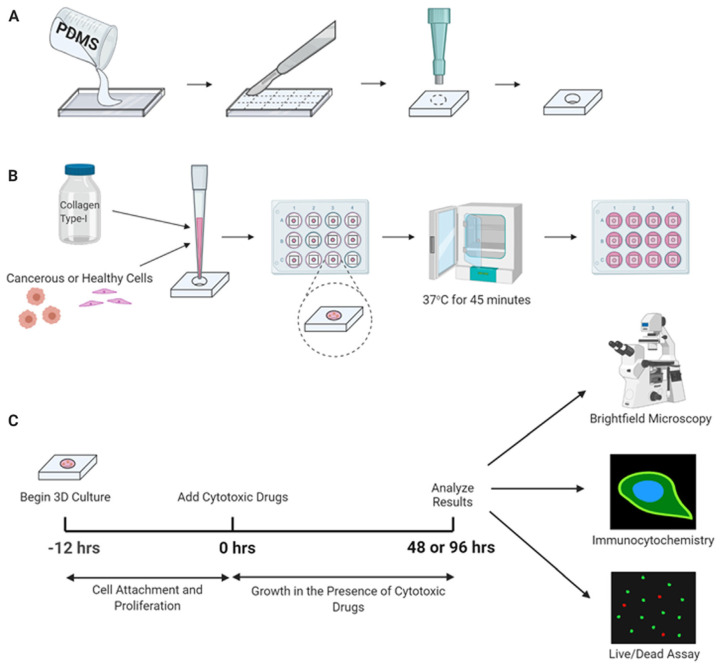
The 3D TEM fabrication workflow for RMS in in vitro studies: (**A**) PDMS mold fabrication using biopsy punch, (**B**) thermal crosslinking of collagen hydrogel mixture with cells are conducted in 12-well plates for 45 min at 37 °C before cell culturing, (**C**) after 3D culturing of constructs, the drugs are added and their cytotoxic effect is analyzed using bright-field microscopy, immunocytochemistry, or live/dead viability assays after 48 or 96 h cell exposure to the cytotoxic drugs. Reprinted from [463].

**Table 2 cancers-15-05269-t002:** RMS targeted therapies and their clinical trial status.

Treatment	Clinical Trial Phase	Reference
Pazopanib	II	[64]
Pazopanib or placebo	III	[65]
Sorafenib	II	[66]
Sorafenib	II	[67]
Crizotinib	II	[68]
Temsirolimus	II	[69]
Cixutumumab	II	[70]
Cixutumumab	II	[71]

**Table 4 cancers-15-05269-t004:** Recent clinical trials on novel therapeutic agents against RMS.

Therapeutic Agents	Clinical Trial ID	Number of Participants	Study Phase	Comments
Abemaciclib	NCT04238819	60	I	Study recruiting
Temsirolimus or bevacizumab	NCT01222715	87	II	Study completed, has results
Cixutumumab	NCT00668148	113	II	Study completed, has results
Cixutumumab	NCT00831844	116	II	Study completed, has results
Cixutumumab or temozolomide	NCT01055314	175	II	Study completed, has results
Cixutumumab and Ttemsirolimus	NCT01614795	46	II	Study completed, has results
Crizotinib	NCT01524926	582	II	Study active, not recruiting
Onivyde and talazoparib or temozolomide	NCT04901702	160	I/II	Study not yet recruiting
Palbociclib	NCT03709680	133	I	Study recruiting
Pazopanib	NCT01532687	54	II	Study completed, has results
Regorafenib	NCT02048371	150	II	Study recruiting
Regorafenib	NCT02085148	62	I	Study completed, has results
Sorafenib	NCT01502410	20	II	Study completed, has results
Sorafenib	NCT02050919	20	II	Study completed, has results
Temozolomide	NCT01355445	120	II	Study completed, has results
Temsirolimus	NCT02567435	397	III	Study recruiting
Temsirolimus	NCT00106353	71	I and II	Study completed, has results
Temsirolimus	NCT00949325	24	I and II	Study completed, has results
Trabectedin	NCT00070109	50	II	Study completed, has results
Vinorelbine	NCT04994132	100	III	Study not yet recruiting
Vinorelbine	NCT00003234	50	II	Study completed, has results
Vinorelbine	NCT04994132	100	III	Study not yet recruiting

**Table 5 cancers-15-05269-t005:** Genetic and pharmacological inhibition of autophagy synergize with therapeutic agents in RMS.

Model	Therapeutic Agent	Autophagy Inhibitor		Act	Outcomes/Effects	Ref.
		Pharmacologic	Genetic			
ARMS cell lines (RH30 and RH4)	Temozolomide	Bafilomycin A1	-	Inhibition of V-ATPase/ATG7	Promoted chemotherapy efficacy	[250]
Human RMS cell line (hRD)	Doxorubicin	Simvastatin	-	Activation of mitochondrial apoptotic pathway (BAX)	Improved the sensitivity of cancer cells towards Dox and improved antitumor activity	[219]
ERMS CSC cell lines	Doxorubicin	Omeprazole	V_0_c siRNA	Inhibition of V-ATPase/lysosomal pH	Enhanced cytotoxic effect of chemotherapy and reduced the invasive potential of ERMS CSCs	[290]
Human RMS cell lines (RH30 and hRD)	Ciclopirox Olamine	Chloroquine	-	Inhibition of lysosomal pH	Improved antitumor activity	[291]
Human RMS cell lines (RH30 and hRD)	Bortezomib and 17-DMAG	Chloroquine	-	Inhibition of lysosomal pH/UPS and HSR systems	Enhanced drug-induced apoptosis	[292]
ERMS (RD) and ARMS (RMS13) cell lines	Bortezomib	Bafilomycin A1/ST80	BAG3 siRNA	Inhibition of V-ATPase/ATG7	Impaired cancer cell growth and increased cell death	[293]
ERMS cell lines (RD, RH30 and RMS)	Tenovin-6	-	SIRT1 and SIRT2 siRNA	Inhibition of Sirtuins	Impaired cancer cell growth and increased apoptosis	[301]
Human RMS cell line (hRD)	Methotrexate and SDH	-	-	Inhibition of P-gp	Enhanced methotrexate-mediated cytotoxicity	[328]
ARMS RH30 (FG+)	Vincristine	Etoposide	-	Inhibition of PLK1/activation of mitochondrial apoptotic pathway (BAX/BAK)	Improved antitumor activity and increased apoptosis	[332]
ARMS RH30 and ERMS RD, TE381.T (FG+)	Vincristine	Volasertib	-			
ARMS RH30 (FG+)	Doxorubicin	Etoposide	-			
ARMS RH30 (FG+)	Eribulin	Etoposide	-			[333]
ARMS RMS1 (FG+)	Etoposide	Volasertib	-			[331]

ARMS: alveolar rhabdomyosarcoma, V-ATPase: vacuolar H+ ATPase, ERMS: embryonal rhabdomyosarcoma, siRNA: small interfering RNA, V_0_c: V0c ATPase, Dox: doxorubicin, CSC: cancer stem cells, 17-DMAG: 17-(dimethylaminoethylamino)-17-demethoxygeldanamycin, UPS: the ubiquitin-proteasome system, HSR: the heat shock response, ST80: The cytoplasmic histone deacetylase 6 inhibitor ST80, BAG3: Bcl-2-associated athanogene 3, PQC: protein quality control system, SDH: silibinin di-hemisuccinate, P-gp: P-glycoprotein, FG +: PAX3-FOXO1 fusion genes positive, PLK1: polo-like kinase 1.

**Table 6 cancers-15-05269-t006:** Studies for KRAS-induced RMS in zebrafish.

Parameter	Method	Tumor Onset	Outcomes	Ref
*HDAC6*	*CRISPR/Cas9 method for deletion of HDAC6*, constructs containing rag2-KRAS^G12D^-U6-hdac6* guide RNA, *rag2-Cas9* and *myogenin-H2B-RFP injected in 1st-cell stage*	15–20 days of post-fertilization	HDAC6 was found to have significant role in ERMS tumorigenesis, promoting tumor growth, metastasis, and self-renewal.	[371]
*RAC1*	Engraftment of *KRAS*-driven zebrafish ERMS tumors co-expressing GFP and mutant *RAC1*** (*RAC1V12*), dorsal subcutaneous way	Tumor harvest after 3 weeks	Zebrafish expressing *RAC1V12* exhibited more aggressive tumor growth and invasiveness compared to the control group (empty vector).	[371]
*tp53*	*KRAS*-induced ERMS generated in *tp53^del/del^* zebrafish	Tumors were tracked 90 days	Deletion of *tp53* increased metastasis and invasion of ERMS cells, but not the total frequency of tumor cells.	[372]
*Van Gogh-like 2 (Vangl2)*	*KRAS*-induced ERMS is generated in fish with additional *Vangl2* gene	15 days of post-fertilization, 90 days after transplantation	Expression of *Vangl2* supports TPCs and has positive effect for their self-renewal. No effect of *Vangl2* was found on the size, penetrability, and latency of the ERMS tumors.	[373]
*Intracellular NOTCH1 (ICN1)*	*KRAS*-induced ERMS (*KRAS^G12D^* and *KRAS^G12D^-ICN1*) was generated in transgenic zebrafish expressing *myf5-GFP* and *mylz2-mCherry*	Tumors imaged over 100 days after transplantation to the recipient fish	*ICN1* enhanced the number of tumor-propagating cells in zebrafish ERMS, by blocking the differentiation of zebrafish ERMS cells into self-renewing *myf5* positive TPCs.	[374]
*myf5*	*KRAS*-induced ERMS was generated in zebrafish with *rag2-KRAS^G12D^*, with additional *mylpfa-mCherry, myf5-GFP* injection	Animals were imaged after 35 days post-fertilization	Re-expression of *myf5* enhanced tumor formation and penetration, thus had a role in reprogramming of ERMS cells into TPCs.	[375]
GSK3 inhibitors screening	KRAS-induced ERMS in *myf5-GFP* and/or *mylz2-mCherry* transgenic fish	Tumor engraftment was monitored from 10 to 120 days after drug treatment	GSK3*** inhibitors suppressed ERMS growth, depleted TPCs, and blocked self-renewal while activating the WNT/β-catenin pathway.	[376]
Screening of PD98059 and TPCK drugs	*rag2-KRAS^G12D^* and *rag2-DsRed* transgenic zebrafish	Tumors were observed after 7–10 days of post-fertilization	Tumor growth was reduced with the drug treatments, showing anticancer potential.	[377]

HDAC6*: histone deacetylase 6, RAC1**: Ras-related C3 botulinum toxin substrate 1, GSK3***: glycogen synthase kinase 3.

**Table 7 cancers-15-05269-t007:** Available animal model approaches for RMS and comparative evaluation.

Animal Model	Injection Types	Pros	Cons
CDX	Heterotopic (subcutaneous) engraftment—Easy to apply and used to monitor tumor growth. In therapeutic applications, drug response may differ from the orthotopic engraftment. Orthotopic engraftment—The most preferable injection type for clinical applications due to high prediction value. Technically, this injection technique is challenging and difficult to monitor the tumor growth.	- Easy to scale - Low cost and high availability - Easy to manipulate	- Low yield in tumor tissue observation - Therapeutic applications are limited - Low clinical relevance
PDX		- High feasibility and good tumor reflection - Strong clinical relevance - High therapeutic prediction	- High cost and prolonged time are required - Therapeutic applications are limited - Low availability
EIMM		- Suitable for tumor initiation and progression observation - Cost is moderate - Feasibility is moderate	- Low clinical relevance depending on the sarcoma type
GEMM		- Easy to manipulate the expression of genes - Wide variety of applications (i.e., tumorigenesis, tumor progression, and maintenance) - High therapeutic prediction	- High cost - Difficult feasibility - Low availability for the rare type of sarcomas

**Table 8 cancers-15-05269-t008:** The specifications of phantom production methods.

Phantom Production Method	Advantages	Disadvantages
FFF	Simple Wide range of materials Multiple compositions of materials Good mechanical properties	Expensive and low accuracy Supports are required Shear stress on nozzle tip wall Sintering is required in some case
SLA	Speed, pieces can be manufactured within hrs or a day Optimal mechanical features, so they can resist machining Good surface finish Complex geometries	Expensive technology Photopolymers are sticky and messy Printed parts need to be cured Supports are required
Material jetting	High resolution Compatible with a wide range of viscosities High accuracy	Low printing speed High cost

## Data Availability

Not applicable.

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
