# Peer review of "Rhabdomyosarcoma: Current Therapy, Challenges, and Future Approaches to Treatment Strategies"

_cancers, 2023, doi:10.3390/cancers15215269_

Round 1

Reviewer 1 Report

Comments and Suggestions for Authors

This a very long review and serves as a good assessment of available treatment strategies. Unfortunately I am not high expert in medications / chemotherapy and the related experimental models,  but probably know more of chemotherapy - more at least than an average pediatric surgeon - and this part the authors have done thoroughly. 

From the part of therapy and challenges the authors have mostly omitted radiation therapy and indications of local radiation therapy ie. brachytherapy that is applicable in a number of cases currently. 

Of surgery and the challenges there is very little - a classical view of ´resection with margins´  and ´non-mutilating ´  and  ´re-excisions´which is rather a limited view and  not in par with the high quality of the rest of the review . The challenges of RMS surgery include decisions whether or not extensive operations vs local excisions and whether or not sacrifice normal functions or risk local recurrence by too limited surgery . These problems are evident in prostate / bladder or perineal / perianal lesions. 

You should also comment on the possibilities and challenges of intra-operative assessment of extent  of surgical excisions - frozen sections / indocyanine green methods. 

The authors have also omitted long term sequelae of treatment and current possibilities of replacement of joints / bones and survival and quality of life after extensive resections ( urinary diversion, fecal diversion prostheses). 

I think it would improve the review if you could assess these issues in 2 pages - a fraction of the pages for the rets of issues. 

Author Response

Reviewer #1

Comments and Suggestions for Authors

This a very long review and serves as a good assessment of available treatment strategies. Unfortunately I am not high expert in medications / chemotherapy and the related experimental models,  but probably know more of chemotherapy - more at least than an average pediatric surgeon - and this part the authors have done thoroughly. 

From the part of therapy and challenges the authors have mostly omitted radiation therapy and indications of local radiation therapy ie. brachytherapy that is applicable in a number of cases currently. 

Of surgery and the challenges there is very little - a classical view of ´resection with margins´  and ´non-mutilating ´  and  ´re-excisions´which is rather a limited view and  not in par with the high quality of the rest of the review . The challenges of RMS surgery include decisions whether or not extensive operations vs local excisions and whether or not sacrifice normal functions or risk local recurrence by too limited surgery. These problems are evident in prostate / bladder or perineal / perianal lesions. 

You should also comment on the possibilities and challenges of intra-operative assessment of extent of surgical excisions - frozen sections / indocyanine green methods. 

The authors have also omitted long term sequelae of treatment and current possibilities of replacement of joints / bones and survival and quality of life after extensive resections ( urinary diversion, fecal diversion prostheses). 

I think it would improve the review if you could assess these issues in 2 pages - a fraction of the pages for the rets of issues. 

Reply: The authors acknowledge the careful consideration of the respected reviewer on our article. The valuable clinical feedback of the respected reviewer is very important and applying them improve the quality of the paper. The authors have done their best to address the reviewer’s feedback. We added a more detailed section for surgical approaches for RMS treatment and tried to cover the respected reviewer comment (219-288). For the radiotherapy part, we did not add more details as our review is already too long and adding more details make the paper very exhaustive.  

Reviewer 2 Report

Comments and Suggestions for Authors

Zarrabi et al., provide a review on current therapy, challenges, and future approaches to treatment strategiesof rhabdomyosarcomaThis review is very detailed but does not seem very focused to the reviewer and could definitely be improved here.

For example, it does not really seem necessary to present the mode of action of 1st line chemotherapeutic agents. However, their role in apoptosis or influencing autophagy is very important.

3D tissue engineering should be focused on results to improve therapeutic models. Results that improve tissue reconstitution after therapy are significant but less relevant in the context of therapy improvement here.

The authors summarize their review very well in the conclusion and perspectives. This should serve as a template for you to revise and focus your review.

Minor: Figure 4 is missing. Table 7 should be listed before Table 6.

Author Response

Reviewer #2

Comments and Suggestions for Authors

Zarrabi et al., provide a review on current therapy, challenges, and future approaches to treatment strategies of rhabdomyosarcoma. This review is very detailed but does not seem very focused to the reviewer and could definitely be improved here.

For example, it does not really seem necessary to present the mode of action of 1st line chemotherapeutic agents. However, their role in apoptosis or influencing autophagy is very important.

Reply: The authors appreciate the positive feedback of the respected reviewer. We have summarized the mode of action of 1st line chemotherapeutic agents (line 334-496) and then added further details about the role of autophagy in chemotherapy strategies for RMS treatment (767-909).

3D tissue engineering should be focused on results to improve therapeutic models. Results that improve tissue reconstitution after therapy are significant but less relevant in the context of therapy improvement here.

Reply: Thank you very much for your constructive comment. We added the following part for further explanation of 3D tissue engineering to improve the therapeutic model.

Page 41, Lines 1471-1502

“Indeed, the application of 3D printing technology offers numerous advantages over traditional manufacturing methods in different parts of biomedical field. For instance, in the case of tissue engineering, 3D printing employs biocompatible materials like hydrogels and bio-inks to create intricate tissue constructs that replicate native tissue architecture. These constructs could then be used as scaffolds for cell attachment, proliferation, and differentiation, and so fabricate functional tissues and organs that could be transplanted inside the body or could be used as a disease model. Furthermore, 3D printing has revolutionized drug delivery systems, allowing precise control over drug carrier characteristics. This technology has wide applications in the pharmaceutical industry, offering benefits like enhanced efficiency, complex drug release profiles, multiple dosing options, cost-effectiveness, and personalized drug delivery. Personalized medicine tailors drug delivery systems to individual patient needs, considering factors such as age, weight, organ function, and disease severity. 3D printing at the point-of-care facilitates the creation of customized drug combinations, speeding up drug development and reducing costs. Additionally, 3D printing advances high-throughput drug testing with 3D bioprinted tissue models, enabling early-stage biomolecule screening, saving time and resources. The integration of 3D cell culture models with various technologies allows for a more accurate representation of human biology, revolutionizing the understanding of cellular and molecular pathways underlying human diseases and offering innovative solutions to complex healthcare challenges. By incorporating in vitro 3D cell culture models alongside cell lines, significant advancements have emerged, including the development of innovative technologies such as microfluidic devices, tissue-on-a-chip, and organ-on-a-chip systems. These innovations have reduced the limitations and drawbacks related to the application of 2D cell culture and animal models (432-435). 

The performance of this new technique was enhanced by introducing 4D printing method, an emerging and cutting-edge technology in which the dimension of time is incorporated into the printed objects. This technique leads to producing materials with the ability of changing their shape or structure during the implantation, that could promote seamless integration with adjacent tissue, and foster optimal cell proliferation and differentiation. Besides, they present an avenue for the creation of adaptive drug delivery systems, designed to release therapeutic payloads in response to precise physiological triggers, such as variations in pH or temperature, ensuring targeted and controlled drug dispersion (436).”

We also corrected some other parts in this section to improve the article based on the respected reviewer feedback which are as follows:

 Page 42, Lines 1529-1533

“Indeed, the presence of collagen in the structure of this scaffold induced the production of some biochemical cues that led to the attachment and growth of cells, from one side, and aligned the physically designed topography of the cells, from the other side. One of the main targets of fabricating 3D printed scaffolds is creating a microenvironment with the most similarity to the native extracellular matrices.”

Page 42, Lines 1538-1545

“It was a stable scaffold with microporous structure with proteolytic degradable property, that showed long-term culturing ability. The microporous structure of the scaffold and the protein adsorption capability of GO provide the capability of cell attachment, proliferation, and normal metabolism. Besides, GO could induce myogenesis due to its interesting features such as its electrical conductivity, roughness, and the surface oxygen contents. Scaffolds could also be prepared using biomimetic methods that have the ability of replicating the native tissue microenvironment in terms of parallel-aligned structures and the incorporation of biophysical signals.”

The authors summarize their review very well in the conclusion and perspectives. This should serve as a template for you to revise and focus your review.

Minor: Figure 4 is missing. Table 7 should be listed before Table 6.

Reply: Thank you so much for your comment, Figure 4 is added to the main text, Table 7 was listed in line 1105.

Round 2

Reviewer 1 Report

Comments and Suggestions for Authors

I am satisfied with editions

Reviewer 2 Report

Comments and Suggestions for Authors

accept